# The latency-associated transcript locus of herpes simplex virus 1 is a virulence determinant in human skin

Emilia A. H. Vanni[1]*, Joseph W. Foley[2], Andrew J. Davison[3], Marvin Sommer[1], Dongmei Liu[4], Phillip Sung[1], Jennifer Moffat[4], Leigh Zerboni[1], Ann M. Arvin[1]

1 Departments of Pediatrics and Microbiology and Immunology, Stanford University School of Medicine, Stanford, California, United States of America, 2 Department of Pathology, Stanford University School of Medicine, Stanford, California, United States of America, 3 MRC-University of Glasgow Centre for Virus Research, Glasgow, United Kingdom, 4 Department of Microbiology and Immunology, State University of New York-Upstate Medical University, Syracuse, New York, United States of America

* emiliav@stanford.edu

**Data Availability Statement:** The DNA and RNA sequencing data generated in this study have been submitted to the NCBI BioProject database (https://www.ncbi.nlm.nih.gov/bioproject) under accession

## Abstract

Herpes simplex virus 1 (HSV-1) infects skin and mucosal epithelial cells and then travels along axons to establish latency in the neurones of sensory ganglia. Although viral gene expression is restricted during latency, the latency-associated transcript (LAT) locus encodes many RNAs, including a 2 kb intron known as the hallmark of HSV-1 latency. Here, we studied HSV-1 infection and the role of the LAT locus in human skin xenografts *in vivo* and in cultured explants. We sequenced the genomes of our stock of HSV-1 strain 17s*yn*⁺ and seven derived viruses and found nonsynonymous mutations in many viral proteins that had no impact on skin infection. In contrast, deletions in the LAT locus severely impaired HSV-1 replication and lesion formation in skin. However, skin replication was not affected by impaired intron splicing. Moreover, although the LAT locus has been implicated in regulating gene expression in neurones, we observed only small changes in transcript levels that were unrelated to the growth defect in skin, suggesting that its functions in skin may be different from those in neurones. Thus, although the LAT locus was previously thought to be dispensable for lytic infection, we show that it is a determinant of HSV-1 virulence during lytic infection of human skin.

## Author summary

Herpes simplex virus type 1 (HSV-1) infects and destroys the outer layer of skin cells, producing lesions known as cold sores. Although these lesions heal, the virus persists in the host for the lifetime and can reactivate to cause new lesions. This is possible because the virus enters the axons of neurones in the skin and moves to their cell bodies located in spinal or cranial nerve bundles called ganglia, where the virus becomes dormant (latent). The most abundant viral RNAs expressed during this state are the latency associated transcripts (LATs), which have been considered a hallmark of HSV-1 latency. Here, we studied HSV-1 infection and spread in human skin. Unexpectedly, we found that the LAT

numbers PRJNA338711 and PRJNA544722 (17dmiR-H1/H6 genome sequencing only).

**Funding:** This work was supported by the National Institute of Allergy and Infectious Diseases (NIAID; www.niaid.nih.gov/) (AI116857 granted to A.M.A.). The genome sequence analyses were supported by Medical Research Council (https://mrc.ukri.org/) grant MC_UU_12014/3 to A.J.D. J.M. is supported by an NIAID contract HHSN27220100030I. Stanford Cell Sciences Imaging Facility is supported, in part, by Award Number S10RR02557401 from the National Center for Research Resources (NCRR; https://orip.nih.gov/). The funders had no role in study design, data collection and analysis, decision to publish, or preparation of the manuscript.

**Competing interests:** The authors have declared that no competing interests exist.

locus is necessary for lesion formation in skin. HSV-1 viruses that were genetically mutated to delete the start of the locus could not spread in skin, whereas viruses with many other genetic mutations had this capacity. Our results suggest that an antiviral drug that inhibits transcripts from this region of the viral genome could block viral spread in skin, or a vaccine could possibly be produced by genetically modifying the virus at the LAT locus and by doing so, limit the virus' ability become latent in neurones.

## Introduction

Herpes simplex virus 1 (HSV-1) infects mucocutaneous epithelial cells. By entering nearby sensory ganglia nerve termini, it is then able to travel in retrograde fashion to the cell bodies to establish latency. Reactivation and anterograde transport back to epithelia ensure HSV-1 transmission. Although productive infection and genome silencing in neurones have been investigated in cell culture and animal models, little is known about the molecular requirements for HSV-1 infection in differentiated human cells within epithelial tissue *in vivo*. Furthermore, many HSV-1 transcripts are expressed in a cell-type specific manner, and their functions may be related to the cells in which they are expressed [1].

The 152 kbp HSV-1 genome takes the form $TR_L$-$U_L$-$IR_L$-$IR_S$-$U_S$-$TR_S$, where $U_L$ and $U_S$ are unique regions flanked by inverted repeats $TR_L/IR_L$ and $TR_S/IR_S$, respectively. The latency-associated transcript (LAT) locus is located in $TR_L/IR_L$ and thus is present in two copies in the genome. The LAT locus has been considered to be uniquely related to HSV-1 latency, as it is abundantly transcribed in human, mouse, and rabbit neurones, whereas other viral genes are largely silenced [1–4]. The LAT promoter has binding sites for cell transcription factors and a downstream long-term enhancer region that includes a proposed second promoter [1,5–8]. Transcription from the LAT locus is complex due to the number of transcripts encoded, which are collectively known as LATs (**Fig 1**). They include an 8.3 kb primary LAT, which is spliced to yield a 2.0 kb stable lariat intron (**Fig 1**) that is further processed to a 1.5 kb intron in neurones [9–13]. Other LAT locus transcripts include upstream of LAT (UOL) [14]; a 0.7 kb transcript [15]; α0 intron 1 RNAs [16]; L/STs, which include open reading frames (ORFs) O and P [17,18]; several miRNAs (miR-H1 to H5 and H7) [19–21]; and small transcripts sRNA1 and 2 [22] (**Fig 1**). LAT is anti-sense to the RL2, RL1, and RS1 genes, which encode the ICP0, ICP34.5, and ICP4 proteins (**Fig 1**). The AL-RNA, AL-2, and AL-3 transcripts [23,24]; and several miRNAs (miR-H6, H14, H15, H17, and H27) [19,20,25] are anti-sense to LAT (**Fig 1**).

Previous studies in animal models showed that disrupting the LAT locus promoter impairs both establishment of and reactivation from latency [1,26–30]. The LAT locus is also implicated in regulation of viral transcription [31–33] and inhibition of apoptosis [27,34] in neurones. LAT miRNAs have been shown to target specific viral transcripts, thus contributing to gene silencing [20]. However, other mechanisms for these LAT-associated functions remain unknown. In general, LATs are considered to be functional only in neurones since the locus appears to be unimportant for peripheral infection in animal models or replication in most cultured cells [30,31,33]. An exception is the LAT-negative mutant 17ΔN/H, which exhibited decreased viral release and a small plaque phenotype in simian CV-1 cells [28]. To our knowledge, no studies of LAT mutant viruses in human epithelial tissue have been reported.

Here, we studied HSV-1 pathogenesis using human skin xenografts in severe combined immunodeficiency (SCID) mice. This approach allows the ability of viruses to replicate and spread to be tested under conditions mimicking primary infection in the natural host. In skin xenografts, HSV-1 strain 17*syn*$^+$ formed cutaneous lesions that were typical of clinical HSV-1

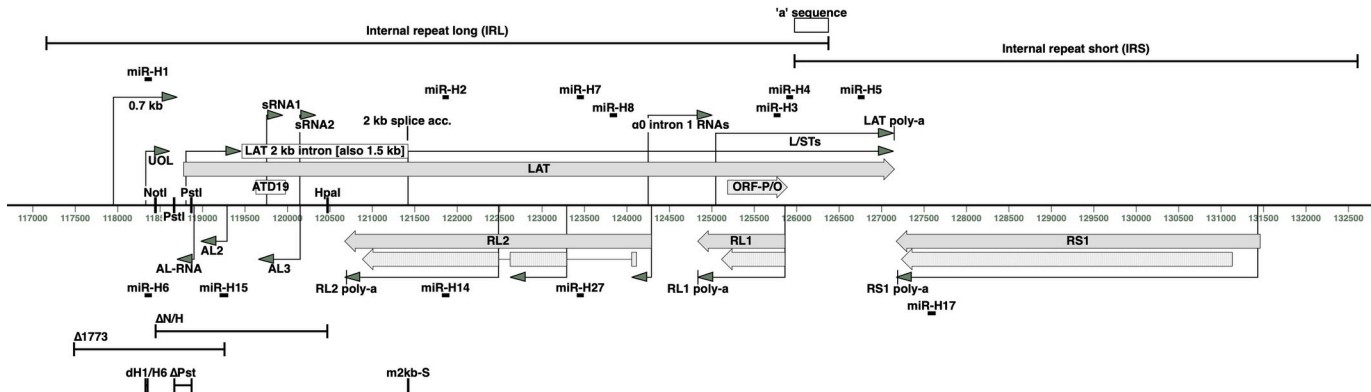

**Fig 1. Schematic of the HSV-1 LAT locus RNAs.** The LAT locus is duplicated in the TR_L and IR_L inverted repeats of the HSV-1 genome, here showing IR_L next to IR_S that together span 117940:127160 in HSV-1 strain 17$syn^+$ genome (GenBank accession JN555585.1). The latency-associated transcript promoter LAP1 is followed by an enhancer element LAP2. The primary 8.3 kb LAT transcript has two exons and a 2.0 kb intron. These, together with multiple other RNAs (0.7 kb, UOL, sRNA1 and 2, α0 intron 1 RNAs 1–4, and L/STs giving rise to the ORF-P and ORF-O proteins) and miRNAs (miR-H1, H2, H7, H8, H3, H4, and H5) transcribed from within or partially overlapping the LAT locus are known as LATs. The RL2 and RL1 genes (yielding the ICP0 and ICP34.5 proteins) are located anti-sense to LAT. In addition, many RNAs (AL-RNA, AL2, and AL3) and miRNAs (miR-H6, H15, H14, and H27) have been reported to map anti-sense to the LAT locus. The RS1 gene (ICP4 protein) and miR-H17 are on the opposite strand in IR_S. The LAT, RS1, RL1, and RL2 polyadenylation sites are indicated. The LAT probe ATD19 binding site is indicated. Changes in five LAT mutant viruses used in the study are marked. The 17ΔN/H deletion spans NotI-HpaI (118447:120472), containing LAP1, exon 1, and part of the 2 kb intron. The Δ1773 deletion removes sequences prior to LAP1 and part of exon 1 (117486:119258). 17dmiR-H1/H6 (dH1/H6) has a 25 nt deletion removing both miR-H1-5p and miR-H6-3p (118328:118352). m2kb-S contains three mutations at the LAT 2 kb intron splice acceptor site (at 121420, 121421, and 121424 nt). The KOSΔPstLAT deletion spans PstI-PstI (118558:118760).

infections, and LAT locus transcription was abundant. We hypothesised that the LAT locus may be critical for pathogenesis when evaluated in fully differentiated skin tissue. To test this, we used 17$syn^+$-derived viruses, including LAT mutants that were previously evaluated in neurotropism models, and sequenced their genomes to link the pathogenesis findings to mutations present in each virus. We found variations in the genomes that affected proteins of all kinetic classes but had no impact on skin virulence. In marked contrast, deleting the 5′ end of the LAT locus severely impaired pathogenesis of HSV-1 in skin. Contrary to observations in neurones [33], this type of deletion did not influence transcription of viral protein-coding genes in human skin. The 2 kb LAT intron was spliced in infected skin, as in neurones, but mutating its splice acceptor site did not alter skin replication. Overall, the impact of deleting the 5′ end of the LAT locus on HSV-1 infection of human skin was much more dramatic than has been reported in latent infection models. We conclude that the 5′ end of the LAT locus is a determinant of HSV-1 virulence in human skin. Since the LAT locus is important for both lytic and latent infection, understanding its functions in human skin is relevant for vaccine design and could be used for identifying novel antiviral targets.

## Results

### Viral genome sequencing

Since our goal was to investigate the role of the LAT locus in human skin infection in relation to studies of LATs in latency, the experiments were done using HSV-1 17$syn^+$ and derived mutants that had been evaluated previously in latency models. These included two deletion mutants in the 5′ end of the LAT locus, 17ΔN/H and 17ΔPst, and the rescuant of the latter, 17ΔPstR (**Fig 1**) [28,35]. Although we were able to obtain the 17ΔN/H mutant that had been tested in the latency model, all stocks of the 17ΔN/H rescuant had been discarded by the laboratory that made these viruses. We also studied a miRNA mutant virus 17dmiR-H1/H6, which contains a deletion encompassing miR-H1-5p and miR-H6-3p located in the vicinity of the

LAP1 promoter [36], and its rescuant 17REdmiR-H1/H6, as the deletions in viruses such as 17ΔN/H may lack parts of the pri-miRNAs (Fig 1). In addition, we used the 17-37 bacterial artificial chromosome (BAC) [37] to make the new mutant m2kb-S, which has substitutions at three nucleotides within the 2 kb LAT intron splice acceptor site to inhibit LAT splicing (Figs 1 and S1). m2kb-S was compared to the parent 17-37 virus generated directly from 17-37 BAC.

Given the lack of a 17ΔN/H rescuant and the inclusion of the previously uncharacterised mutant m2kb-S, all viral genomes were sequenced. This was extremely important because when compared to the consensus sequence of the reference 17*syn*+ genome (GenBank accession JN555585.1), all eight viruses had single nucleotide (nt) polymorphisms (SNPs) and small insertions or deletions (indels) in subpopulations of genomes (S1 Table), including some that affected viral proteins (S1 Table). The viruses also had differences beyond the LAT locus. The genomes of each pair of viruses used in the pathogenesis experiments were compared to each other: 17*syn*+ and 17ΔN/H; 17ΔPst and 17ΔPstR; 17dmiR-H1/H6 and 17REdmiR-H1/H6; and 17-37 and m2kb-S (S1 Table).

## 17*syn*+

Our stock of 17*syn*+ had a total of 43 SNPs/indels compared to the reference sequence, demonstrating a genetic diversity attributable to passage in cell culture and consistent with the original stock having been a mixed population [38]. Of these differences, 26 were present in noncoding regions, and many were related to the lengths of homopolymer tracts. All of the 17 mutations present in ORFs were SNPs, and 10 were nonsynonymous. Based on the proportion of sequence reads, the SNPs affected approximately 14–20% of the genome population except for one in UL44 (54%) and one in UL55 (77%) (S1 Table).

## 17ΔN/H

17ΔN/H was verified to have the expected 2026 nt NotI-HpaI fragment in the 5′ end of the LAT locus (5902:7927 and 118447:120472 in TR_L and IR_L) replaced by a 441 nt sequence from bacteriophage lambda [28]. This deletion removes the LAP1 promoter, the whole of exon 1, and a part of the 2 kb intron (Fig 1). 17ΔN/H had 61 mutations that were located in non-coding sequences or synonymously within ORFs (S1 Table). The virus also had 28 nonsynonymous SNPs (>99%) (S1 Table). In addition, 17ΔN/H had indels (100%) in two C-tracts in US8 (encoding glycoprotein E (gE)) that caused the reading frame to change for a 24 nt region (S1 Table). Three of the 28 synonymous SNPs in 17ΔN/H were also present in 17*syn*+, and 25 were not. Six of those in 17*syn*+ were not found in 17ΔN/H.

## 17ΔPst and 17ΔPstR

17ΔPst was expected to have a 203 nt deletion of a PstI-PstI fragment within the LAT promoter LAP1 (Fig 1) [35]. However, there was a larger 1773 nt deletion in the 5′ end of the LAT locus, which removes the LAP1 promoter and a part of exon 1 (Fig 1). (7116:8888 and 117486:119258 in TR_L and IR_L) (S1 Table). Therefore, we renamed this virus Δ1773. Unexpectedly, it also had replacements of 549 nt within UL29 (61408:61956) and 487 nt within UL30 (65305:65791) with the corresponding HSV-2 sequences (S1 Table). These replacements closely matched the HSV-2 reference sequence (GenBank accession JN561323.2) for UL29 (61836:62384 with two SNPs) and UL30 (65813:66299 with one SNP). UL29 encodes the single-stranded DNA–binding protein, and UL30 encodes the catalytic subunit of DNA polymerase; the origin of DNA replication oriL is located between them [1]. The resulting amino acid sequences were 90 and 96% identical, respectively, to those in HSV-1. The origin of these

sequence substitutions is unknown to us. HSV-2 isolates carrying insertions of HSV-1 UL29 and UL30 sequences causing genital herpes lesions are in global circulation [39,40]. The reverse is possible in HSV-1 genomes, but was not detected in a small sample population [40].

Compared to the 17$syn^+$ reference, Δ1773 also had 28 SNPs/indels in non-coding regions and 28 synonymous SNPs within ORFs. Of the latter, 18 SNPs (>99%) resulted in altered amino acid sequences within 13 different ORFs (S1 Table). 17ΔPstR was thought to be the rescuant of 17ΔPst with a repaired LAP1 LAT promoter [35]. Compared to the 17$syn^+$ reference, the 17ΔPstR sequence revealed 49 SNPs/indels within ORFs, of which 10 SNPs (>94%) and two indels (>96%) altered amino acid sequences within 13 ORFs (S1 Table). No HSV-2 substitution sequences were found in 17ΔPstR. Only one of the nonsynonymous changes in Δ1773 and 17ΔPstR was found in both viruses (S1 Table). Together, these results show that 17ΔPstR was not a rescuant derived from Δ1773.

### 17dmiR-H1/H6 and 17REdmiR-H1/H6

The miRNA mutant 17dmiR-H1/H6 contained a 25 nt deletion (8022:8046 and 118328: 118352 in $TR_L$ and $IR_L$), thus removing both miR-H1-5p (miRbase MIMAT0003744) and miR-H6-3p (MIMAT0008404) [36]. These sequences were intact in 17REdmiR-H1/H6. Compared to the 17$syn^+$ reference, 17dmiR-H1/H6 had 42 SNPs/indels, and 17REdmiR-H1/H6 had 38, most of which were present in one but not the other virus (S1 Table). Only three SNPs (>99%) that altered amino acid sequences were shared between the two viruses, whereas six and seven, respectively, were present in only one virus, indicating that 17REdmiR-H1/H6 was not a direct rescuant of 17dmiR-H1/H6 (S1 Table).

### 17-37 and m2kb-S

Comparison of 17-37 and m2kb-S, which were both generated from 17-37 BAC, showed the expected deletion of a 148 nt region of oriL palindromic region, as well as the anticipated insertion of a 47 nt *loxP* site between UL37 and UL38 due to excision of the BAC vector. The three introduced point mutations at the LAT 2 kb intron splice acceptor site (at 121420, 121421, and 121424 nt in $IR_L$ and the three corresponding nt in $TR_L$) were verified in m2kb-S. These did not change the amino acid sequence of ICP0 encoded on the opposite strand. Compared to the 17$syn^+$ reference, the number of SNPs/indels was 61 in 17–73 and 64 in m2kb-S. Only three of these changes in 17–73 were absent from m2kb-S, and six (including the three introduced point mutations) vice versa. Also, 13 SNPs and one indel (all >83%) led to amino acid sequence changes in both viruses (S1 Table). An indel in UL44 (>99%; one nucleotide deletion in a polynucleotide tract) was present in both viruses (S1 Table) and was predicted to lead to premature termination of glycoprotein C (gC). Indeed, we verified the gC truncation by immunoblotting (S4 Fig). We did not sequence this region in the parental 17-37 BAC but the presence of the same mutation in both progeny viruses suggest that its sequence contains the UL44 indel. Only one SNP in UL6 (8%) and an indel in UL23 (83%) were present in 17-37 but absent from m2kb-S (S1 Table). The high level of similarity between these viruses was consistent with their derivation from the 17-37 BAC by limited passage in cell culture.

In summary, this extensive genomics analysis provided vital information for evaluating subsequent observations about HSV-1 skin pathogenesis and underscored the importance of sequencing the viruses used. For the purpose of understanding the role of the LAT locus in skin infection, all four viruses that did not have deletions in the LAT locus (17$syn^+$, 17ΔPstR, 17REdmiR-H1/H6, and 17-37) served as controls for the LAT mutants (17ΔN/H, Δ1773, 17dmiR-H1/H6, and m2kb-S). In addition, the analysis of the genome sequences of the

17*syn*+, 17ΔPstR, 17REdmiR-H1/H6, and 17-37 viruses lacking LAT mutations made it possible to determine whether the SNPs and indels found in their genomes had effects on HSV-1 infection in skin.

## Deleting the 5′ end of the LAT locus restricts lesion formation in human skin

Typical cytopathology of clinical HSV-1 lesions primarily affects keratinocytes [3,41,42], and lesions develop over approximately 10 days [43]. To study the pathogenesis of HSV-1 in human skin, we used xenografts in SCID mice, which model clinical infection of peripheral sites in the initial phase of tissue infection before adaptive immunity is elicited. This model allows the assessment of viral replication and lesion formation over time in the context of host cell intrinsic defences and innate antiviral responses mediated predominantly by the type I interferon pathway in skin [44].

We compared the ability of 17*syn*+ and a virus lacking the 5′ end of the LAT locus to form lesions, using 17ΔN/H (del 5902:7927 and 118447:120472) for this purpose [28]. Time points between 7–14 days were chosen for analysis because the whole epithelial cell layer becomes depleted after 7 days post-infection (DPI) with 17*syn*+ (**Fig 2**), whereas a later time point was needed to allow for delayed growth of 17ΔN/H. *In situ* hybridisation was used to detect viral DNA and LAT RNA. 17*syn*+ (2,500 plaque-forming units (PFU)) caused progressive destruction of epithelial, but not dermal, cells, and all epithelial cells were positive for viral DNA and LAT RNA at 7 DPI (**Fig 2A**). Infection of hair follicle epithelial cells was prominent. In contrast, xenografts infected with 17ΔN/H (2,500 PFU) caused no apparent damage to the epithelial layer at 11 DPI (**Fig 2B**). Destruction of epithelial cells and viral DNA could be detected at 14 DPI with a very high inoculum of 17ΔN/H (50,000 PFU) (**Fig 2B**). A low inoculum (1,000 PFU) caused lesions in all 17*syn*+-infected xenografts and none of the 17ΔN/H-infected xenografts at 7 DPI. The LAT probe ATD19 binding site is within the ΔN/H deletion and resulted in no signal (**Figs 1 and 2B**).

We used the immediate-early protein ICP0 as an early marker for lesion size and HSV spread, as it is found in the tegument and is the first protein detected in the nucleus upon infection prior to DNA replication [1]. The cellular PML protein was used as a marker, as it is predominantly nuclear. Indeed, ICP0 was expressed in the absence of detectable levels of HSV-1 DNA in 17ΔN/H infected skin at 11 DPI (**Fig 2B and 2D**). ICP0 was present in both 17*syn*+- and 17ΔN/H-infected epithelial cells (**Fig 2C and 2D**). The ICP0 staining also highlighted the inability of 17ΔN/H to form lesions: only a few cells were ICP0-positive at 11 DPI compared to many in 17*syn*+-infected skin at 7 DPI.

To address any experimental bias due to analysing an uninfected skin area rather than a lesion site, we used different markers to detect infection across multiple sequential skin sections. Viral DNA, LAT, or protein expression was identified across 40 adjacent 6 μm sections in 17*syn*+-infected xenografts at 7 DPI (2,500 PFU) (**Fig 2A**), compared to the detection of viral protein in only five adjacent sections of the 17ΔN/H-infected xenograft that yielded the highest viral titre at 11 DPI (**Fig 2B**). These data confirmed that 17ΔN/H lesion formation was severely restricted in human skin.

## HSV-1 replication in human skin xenografts is impaired by deletions in the 5′ end of the LAT locus

We measured the amount of infectious virus in infected skin xenografts to evaluate the replication ability of different viruses, expecting a low yield for 17ΔN/H given its defective capacity to form lesions. At 7 DPI, a high yield of infectious virus was recovered from all eight skin

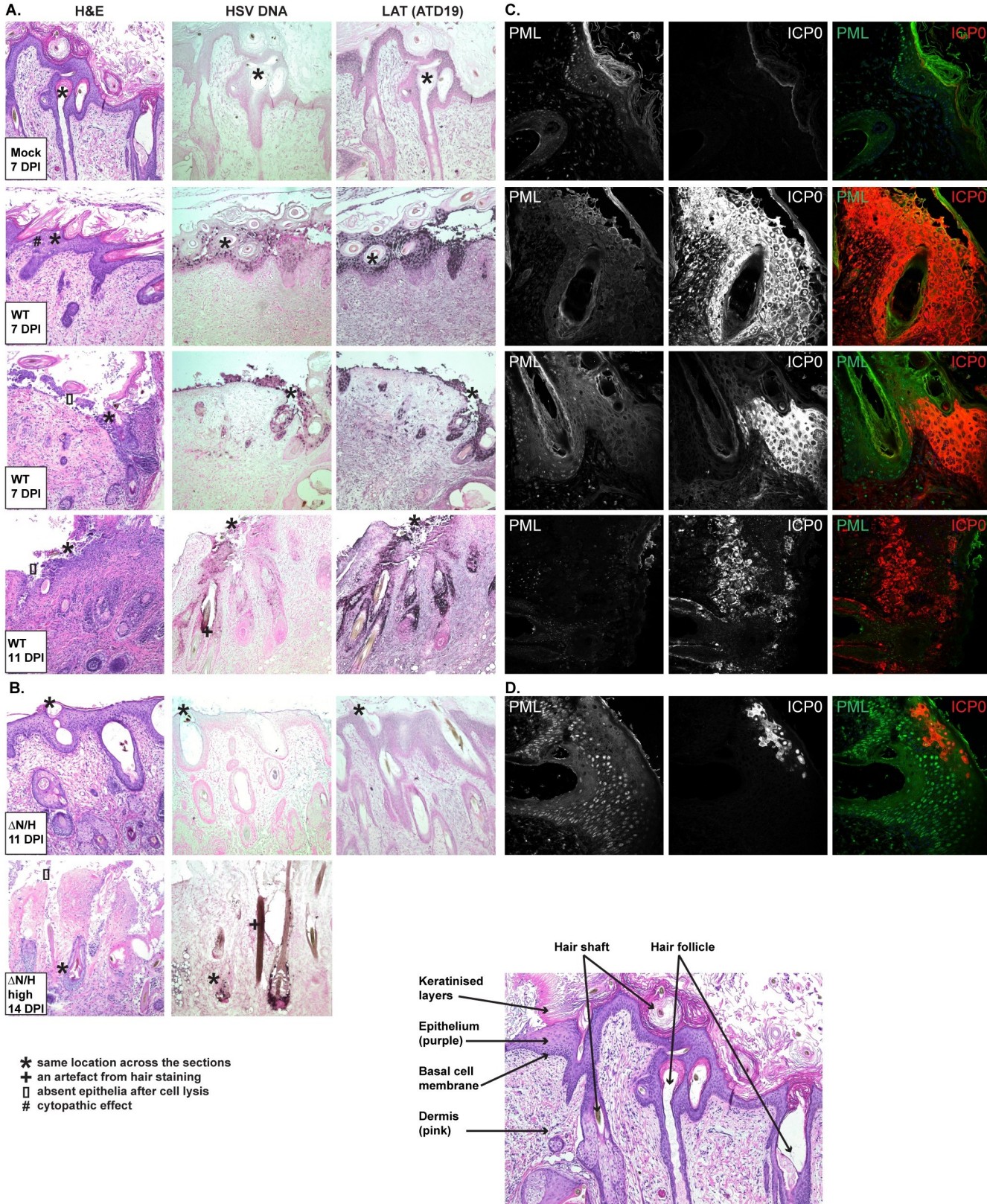

**Fig 2. Cutaneous lesion formation by 17*syn*⁺ and 17ΔN/H in human skin *in vivo*.** (A) Representative mock- and 17*syn*⁺-infected (2,500 PFU) skin sections at 7 or 11 DPI stained with haematoxylin and eosin (H&E; left), and *in situ* hybridisation for HSV-1 DNA (centre) or LAT using ATD19 probe (right). (B)

Representative 17ΔN/H-infected skin sections at 11 DPI (2,500 PFU) or 14 DPI (high indicates 50,000 PFU inocula used to infect the tissue). (C) Representative 17*syn*+ (2,500 PFU) skin sections at 7 or 11 DPI stained for ICP0 and PML. PML was used as a cellular marker. (D) Same staining on a representative 17ΔN/H-infected skin section. A simplified diagram of skin cross-sections is shown (lower right corner). * marks the corresponding location across skin sections, + marks a staining artefact, [] marks an area that used to be covered with epithelial cells, and # marks cytopathic effect.

xenografts infected with 17*syn*+ (1,000 PFU) (mean titre: $10^6$ PFU ± 0.8 SEM). In contrast, as expected, no virus was recovered from any xenografts inoculated with 17ΔN/H (**Fig 3A**). Using a higher inoculum (2,500 PFU), virus was recovered from all 17*syn*+-infected xenografts at 7 and 11 DPI, but only two out of six infected with 17ΔN/H at 7 DPI and three out of six at 11 DPI (**Fig 3B**). Titres in four of the five 17ΔN/H xenografts that yielded virus were lower than those in 17*syn*+-infected tissues (**Fig 3A and 3B**). We further confirmed that the 17ΔN/H growth defect was not rescued at 7 DPI with a very high inoculum (50,000 PFU) (**Fig 3C**). Low viral titres for 17ΔN/H were consistent with small lesions and delayed viral spread (**Figs 2 and 3A–3C**). The high inocula led to high titres at 14 DPI (**Fig 3C**). Some mice had evidence of 17*syn*+ spread to the central nervous system (CNS) at 11–14 days post-infection. 17ΔN/H escape was not noted during the 14-day interval and was attributed to the reduced viral load associated with the impaired replication of this virus in skin xenografts.

Although SCID mice have natural killer cells that may infiltrate skin xenografts over time, we assumed that this response would happen regardless of whether 17*syn*+ or 17ΔN/H was used, and that it would not be the causative factor of the 17ΔN/H growth defect. To address this assumption, we grew skin explants in tissue culture and confirmed that 17ΔN/H had a replication defect in the absence of such cellular immune responses (**Fig 3D**). 17ΔN/H growth was low compared to 17*syn*+ at 4 DPI with an inoculum of 100 or 1,000 PFU (**Fig 3D**). The defect was less pronounced than we observed in the skin xenografts, with both viruses spreading faster (**Fig 3A–3C**). Viral spread is slower in xenografts because they have an intact dermal layer to replenish the epidermis and capillary perfusion to maintain the tissue environment, thus creating more barriers for viral spread, as is the case in the host. In the higher inoculum experiments and those with longer time intervals post-infection, 17*syn*+ titres were lower in both xenograft and *ex vivo* models due to epithelial cell depletion, which left no epithelial cells to support viral replication (**Figs 2 and 3**). Importantly, the replication defect of 17ΔN/H in cultured skin explants mitigates the theoretical concern that the phenotype was due to xenografting methods.

The second LAT mutant, Δ1773, with a 1773 nt deletion in the 5′ end of the LAT locus that does not affect the 2 kb LAT intron sequences (7116:8888 and 117486:119258), was compared to 17ΔPstR in skin xenografts (**Fig 3E**). Like the NotI-HpaI deletion in 17ΔN/H, the 1773 nt deletion caused a growth defect in skin xenografts (**Fig 3E**). The genome sequencing data showed that parts of the UL29 and UL30 gene sequences in Δ1773 had been substituted with equivalent sequences of the HSV-2 UL29 and UL30 genes (**S1 Table**). In human fibroblasts (HFs), we saw no differences between Δ1773 and 17ΔPstR replication (**S2A Fig**). The high amino acid sequence identity (90 and 96%, respectively) makes it unlikely that the substitutions were the cause of the defect in skin pathogenesis. These data supported the conclusion that the defective replication phenotypes observed for 17ΔN/H and Δ1773 in skin were due to the deletions rather than to differences elsewhere in the genome (**S1 Table**).

In addition to the 17*syn*+ mutant viruses 17ΔN/H and Δ1773, we tested KOSΔPstLAT and its rescuant KOSΔPstR [45]. KOSΔPstLAT contains a smaller 203 nt PstI-PstI deletion in the 5′ end of the LAT locus, which removes the basal LAP1 promoter without affecting the sequences encoding the 2 kb LAT intron (7481,7683 and 118558:118760 in GenBank accession JQ673480.1). These strain KOS–derived viruses were not sequenced, but the deletion and its

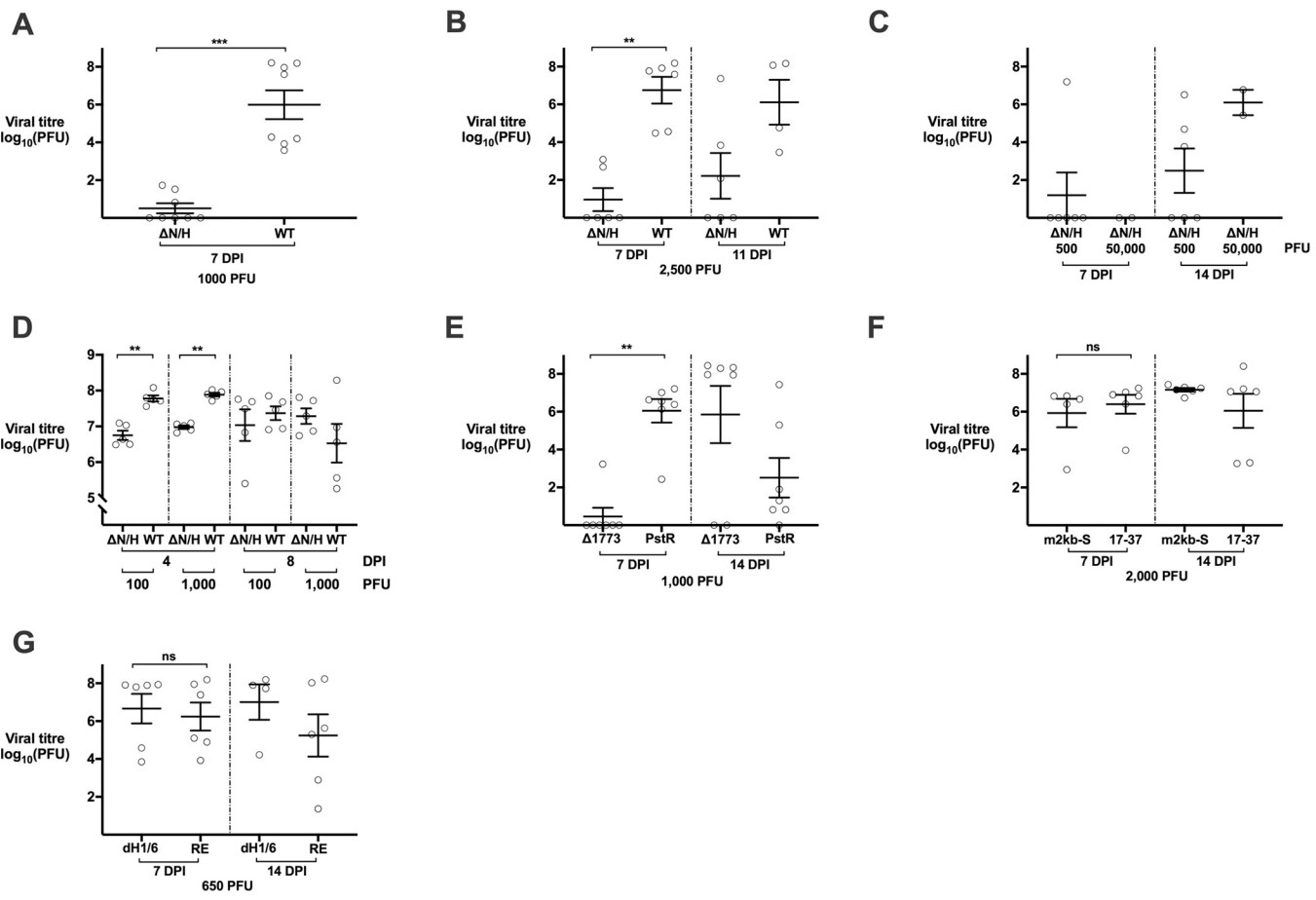

**Fig 3. Replication of 17*syn*+ and LAT mutant viruses in human skin *in vivo* and cultured skin explants.** Viral titres (log₁₀ PFU/implant ± SEM; each data point represents an individual replicate) recovered from: (A) *in vivo* skin xenografts infected with 1,000 PFU 17ΔN/H (ΔN/H) or 17*syn*+ (WT) at 7 DPI (n = 8); (B) *in vivo* skin xenografts infected with 2,500 PFU 17ΔN/H or 17*syn*+ at 7 (n = 6) or 11 DPI (17ΔN/H n = 6; 17*syn*+ n = 4); (C) *in vivo* skin xenografts infected with 500 or 50,000 PFU 17ΔN/H at 7 or 14 DPI (500 PFU n = 6, 50,000 PFU n = 2); (D) cultured skin explants infected with 100 or 1,000 PFU 17ΔN/H or 17*syn*+ at 4 or 8 DPI (n = 5); (E) *in vivo* skin xenografts infected with 1,000 PFU Δ1773 or 17ΔPstR (PstR) at 7 or 14 DPI (n = 7); (F) *in vivo* skin xenografts infected with 2,000 PFU m2kb-S or 17-37 at 7 (m2kb-S n = 5; 17-37 n = 6) and 14 DPI (n = 6); and (G) *in vivo* skin xenografts infected with 650 PFU 17dmiR-H1/H6 (dH1/6) and 17REdmiR-H1/H6 (RE) at 7 (n = 6) and 14 DPI (17dmiR-H1/H6 n = 4; 17REdmiR-H1/H6 n = 6). Mann-Whitney non-parametric unpaired two-tailed tests were performed, for which ns signifies p > 0.05, * p signifies ≤ 0.05, ** signifies p ≤ 0.01, *** signifies p ≤ 0.001, and **** signifies p ≤ 0.0001. The exact p values are (A) *** p = 0.0002 for 17ΔN/H vs. 17*syn*+ at 7 DPI; (B) ** p = 0.0022 for 17ΔN/H vs. 17*syn*+ at 7 DPI; (C) N/A; (D) ** p = 0.0079 for 17ΔN/H vs. 17*syn*+ at 4 DPI (100 PFU inocula) and ** p = 0.0079 for 4 DPI (1000 PFU inocula); (E) ** p = 0.0012 for 1,000 PFU inocula Δ1773 vs. 17ΔPstR at 7 DPI; (F) ns p = 0.1926 for m2kb-S vs. 17-37 at 7 DPI; and (G) ns p = 0.9372 for 17dmiR-H1/H6 vs. 17REdmiR-H1/H6 at 7 DPI.

repair in the rescuant were verified by PCR and sequencing across it. Although the viruses had no differences in ability to replicate in HFs (**S2B Fig**), we noted impaired viral titres for the mutant virus in skin xenografts. Four of five skin xenografts infected with KOSΔPstR had viral titres of >1000 PFU, compared to only one of five infected with KOSΔPstLAT (**S3 Fig**). These data suggest a delayed replication phenotype for KOSΔPstLAT compared to KOSΔPstR, although not as marked as for mutants with larger deletions in the 5′ end of the LAT locus.

### Role of the LAT 2kb intron and LAT locus miRNAs H1 and H6 in HSV-1 skin infection

We generated a new virus, m2kb-S, with a mutated LAT 2 kb intron splice acceptor site, using the 17-37 BAC [37]. We used this virus to assess whether absence of the LAT 2 kb intron

contributed to the impaired pathogenesis in skin observed for viruses lacking the basal LAP1 promoter or larger regions of the 5′ end of the LAT locus (**Figs 3A–3E and S3**). Lack of 2 kb intron splicing in m2kb-S infected cells was verified compared to 17*syn*⁺ and 17-37 by RT-PCR (**S1 Fig**). Although the LAT 2 kb intron was not present, the m2kb-S phenotype was identical to that of 17-37 in skin xenografts (**Fig 3F**) and of the same magnitude as that of 17*syn*⁺ (**Fig 3A and 3B**). Thus, the replication defect observed for the deletion viruses in skin was unlikely to be due to functions of the 2 kb LAT intron.

Since miR-H1 and miR-H6 are located in the LAT locus region (**Fig 1**), 17dmiR-H1/H6 and 17REdmiR-H1/H6 were tested in skin xenografts, using a low inoculum (650 PFU) to allow detection of minor effects on replication [36]. Growth of the 17dmiR-H1/H6 and 17REdmiR-H1/H6 did not differ at 7 or 14 DPI (**Fig 3G**), indicating that these miRNAs within the ΔN/H deletion did not cause the growth deficiency of 17ΔN/H in skin.

## Mutations in many HSV-1 proteins do not alter skin pathogenesis

All four viruses lacking LAT locus mutations (17*syn*⁺, 17ΔPstR, 17REdmiR-H1/H6, and 17-37) retained the capacity to replicate to high viral titres in skin xenografts (**Fig 3**), despite the presence of other genetic differences in various combinations (**S1 Table**). When compared to the 17*syn*⁺ reference sequence, our 17*syn*⁺ virus had only two nonsynonymous substitutions that were present in the majority of genomes; these affected UL44 (54%) and UL55 (77%). The virus caused extensive skin lesions typical of clinical HSV-1 (**Fig 2**). Of the other viruses with an intact LAT locus, 17ΔPstR had 10 amino acid substitutions in UL12, UL14, UL19, UL25, UL29, UL38, RS1, and US8A, and frameshifts in UL46 and RL1 (all >94%) (**S1 Table**). 17-37 had 13 single amino acid substitutions in UL9, UL10, UL22, UL26, UL30, UL32, UL42, UL44, UL54, UL55, UL56, and RS1 (>99%) and two frameshifts in UL23 (83%) and UL44 (100%) (**S1 Table**). 17REdmiR-H1/H6 had 10 single amino acid substitutions in UL11, UL25, UL26, UL34, UL40, UL44, UL55, RL2, US7, and US9 (>98%) (**S1 Table**). Most notably, the frameshift in UL44 in 17-37 and m2kb-S led to truncation of glycoprotein C (>99%) (**S1 Table**) without affecting skin infection. These data indicated that many variations in viral proteins that have critical functions, including envelope fusion, viral gene transactivation, DNA replication, and immune evasion, are compatible with pathogenesis of HSV-1 in epithelial cells *in vivo*.

## LAT locus transcription is dispensable for replication in primary human skin cells in cell culture

Having identified an unexpected growth deficiency in skin *in vivo* for mutants with deletions in the 5′ end of the LAT locus, we asked whether defects had not been found in cell culture because primary human skin cells were not tested. Comparisons of viral titres from supernatant or infected cells showed that 17*syn*⁺ and 17ΔN/H growth did not differ in HFs or primary keratinocytes (MOI 0.5) (**Fig 4A**), indicating that the 17ΔN/H defect in skin was not related to diminished replication capacity. However, using a low inoculum (MOI 0.0001), release of 17ΔN/H from HFs was initially 1 log lower than that of 17*syn*⁺ at 24 hours post-infection (HPI) but similar at 72 HPI, and release from keratinocytes was 2 logs lower than that of 17*syn*⁺ at 24 and 72 HPI (**Fig 4B**). These data support the previous findings of 17ΔN/H exhibiting decreased viral release and a small plaque phenotype in simian CV-1 cells [28]. We could not confirm the plaque phenotype in the absence of the rescuant, and comparison to our 17*syn*⁺ stock would not have been valid because different 17*syn*⁺ stocks have varying plaque phenotypes [38]. Overall, these results demonstrated that infection of primary cells in culture did not reveal the critical role of the 5′ end of the LAT locus for replication in skin.

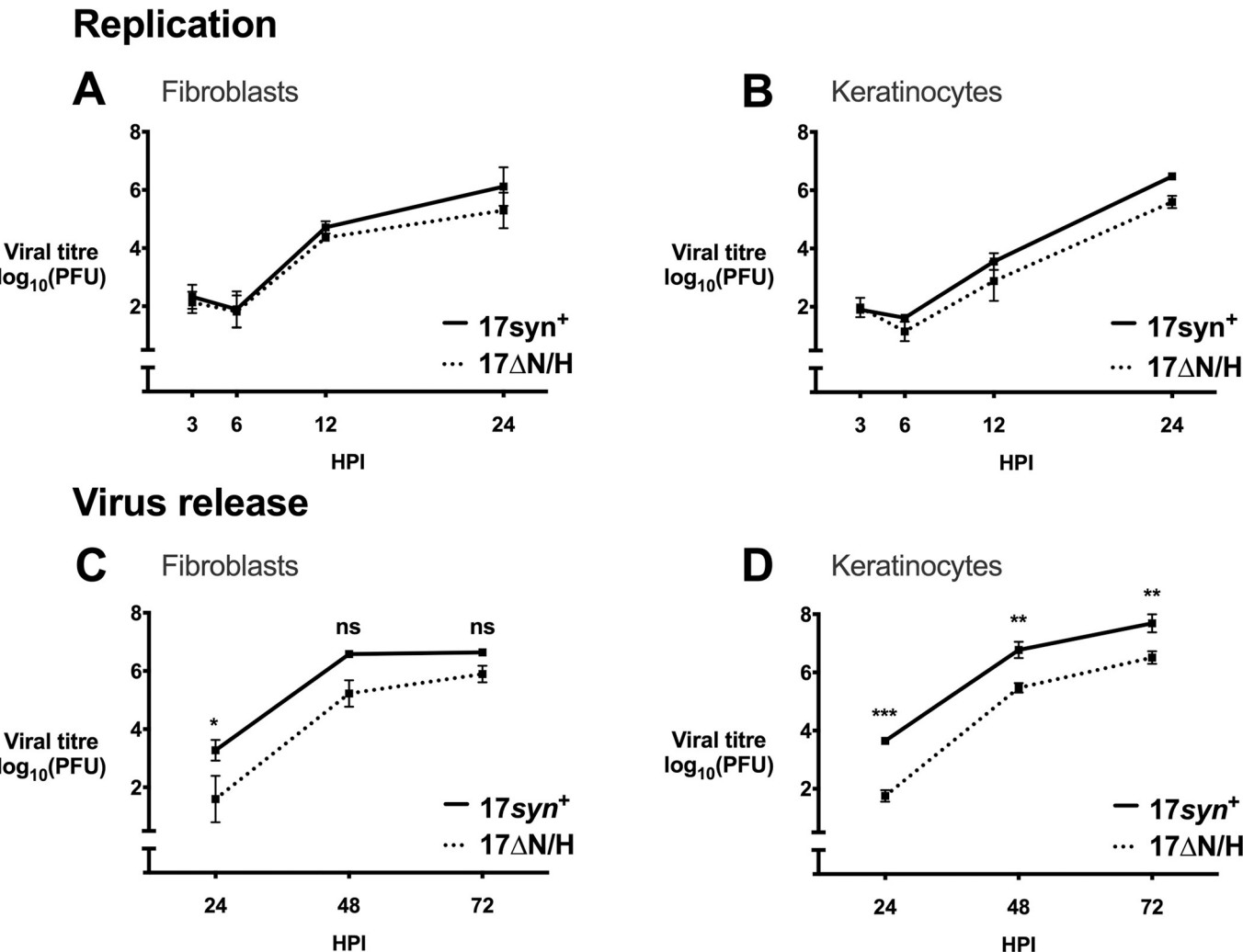

**Fig 4. Replication of 17*syn*+ and 17ΔN/H and virus release in cell culture.** (A) Viral titres (log₁₀ PFU/ml ± SEM) at specified time points from replication assays of 17*syn*+ and 17ΔN/H in HFs or keratinocytes (MOI 0.5; n = 3 for both viruses per time point). 2-way ANOVA was performed with Sidak's correction, where all time points had non-significant P > 0.05 differences. (B) Titres (log₁₀ PFU/ml ± SEM) from 17*syn*+ and 17ΔN/H virus release assays in HFs or keratinocytes (MOI 0.0001; n = 3 for both viruses per time point). 2-way ANOVA was performed with Sidak's correction, where ns signifies p > 0.05, * p signifies ≤ 0.05, ** signifies p ≤ 0.01, *** signifies p ≤ 0.001, and **** signifies p ≤ 0.0001. The exact p values are * p = 0.0498 at 24 HPI, ns p = 0.1265 at 48 HPI, and ns p = 0.5621 at 72 HPI in HFs; and *** p = 0.0002 at 24 HPI, ** p = 0.0041 at 48 HPI, and ** p = 0.0085 at 72 HPI in keratinocytes.

### The 17*syn*+ and 17ΔN/H transcriptomes in human skin

Since LAT has been reported to regulate HSV-1 transcription in neurones [31–33], it was important to determine whether differential HSV-1 transcription might contribute to the defective pathogenesis observed with mutants lacking the 5′ end of the LAT locus in skin. The skin xenograft model provided the opportunity to investigate the HSV-1 transcriptome in epithelial cells infected *in vivo*.

First, we used strand-specific RNA-seq to study the HSV-1 transcriptome in skin. Reads covering all known HSV-1 protein-coding transcripts were detected in both 17*syn*+ and 17ΔN/H-infected skin (2,500 PFU) at 11 DPI (**Fig 5A and 5B and S2 Table**). Since 17ΔN/H replication was limited, reads were normalised against total viral reads and analysed using Salmon [46]. Some viral transcripts had low abundance relative to those with highest coverage in both 17*syn*+ and 17ΔN/H, e.g. RL1 encoding ICP34.5 (**Fig 5A and 5B and S2 Table**).

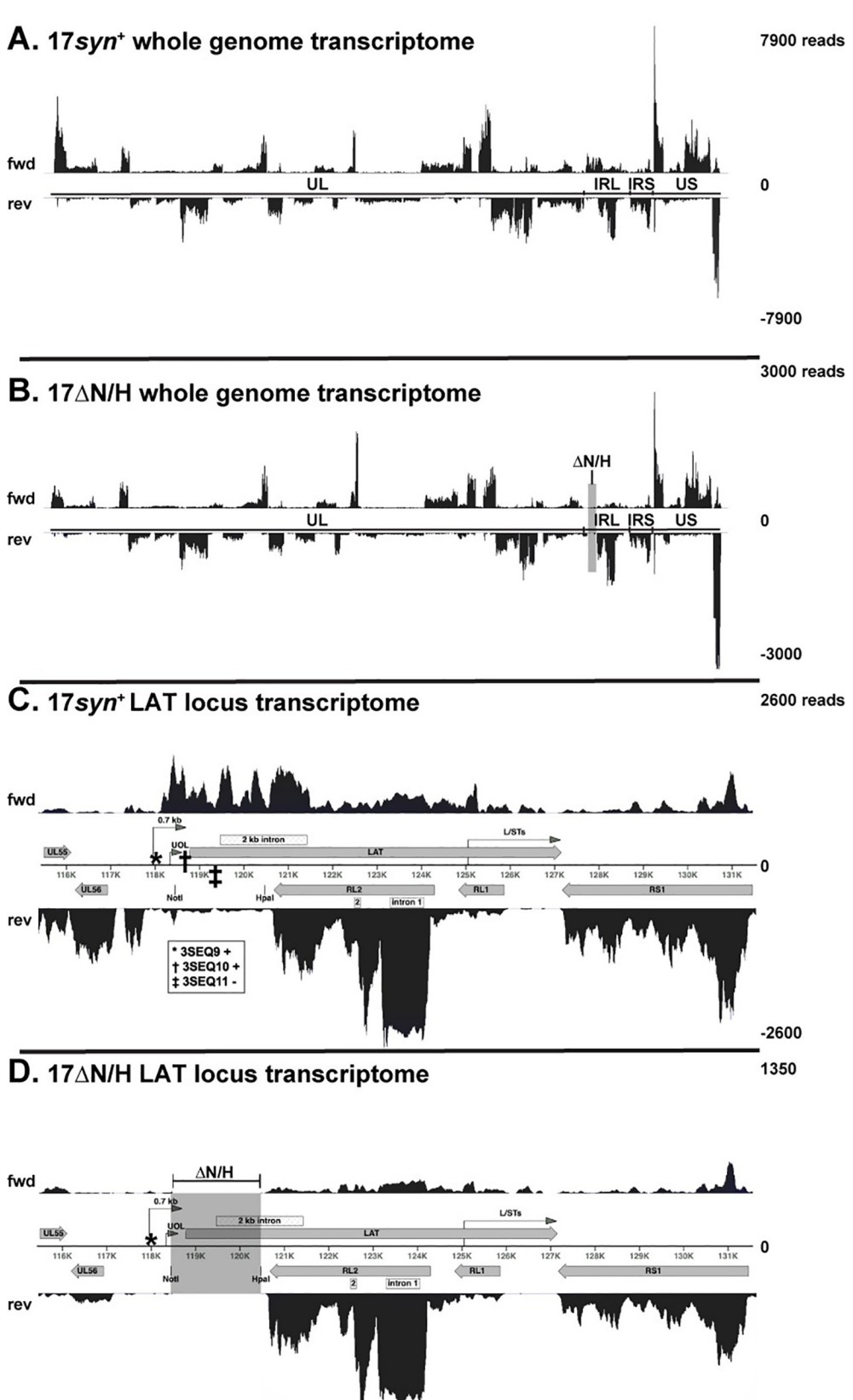

**Fig 5. Viral transcriptome and LAT locus transcription in skin xenografts infected with 17*syn*+ and 17ΔN/H.** Total RNA was extracted from 17*syn*+- or 17ΔN/H-infected (2,500 PFU) skin xenografts at 11 DPI and pooled for RNA-seq. Reads were mapped to a reference genome consisting of $U_L$-$IR_L$-$IR_S$-$U_S$, with the inverted repeats $TR_L$ and $TR_S$ omitted. The indicated read scales were determined by the maximum peak of reads. Histograms of strand-specific (fwd or rev) transcription coverage across (A) 17*syn*+ and (B) 17ΔN/H genomes are shown. The ΔN/H deletion is highlighted by a grey box. A close up of LAT locus transcription is shown for (C) 17*syn*+ and (D) 17ΔN/H. The area shown includes two preceding $U_L$ genes (UL55 and UL56), $IR_L$ genes (LAT, RL1, and RL2), and an $IR_S$ gene (RS1). Additional region hits for polyadenylated transcript ends found using 3SEQ are marked as * 3SEQ9 (fwd), † 3SEQ10 (fwd), and ‡ 3SEQ11 (rev) (which are not annotated in GenBank accession JN555585.1). Two of these, 3SEQ10 and 3SEQ11, are within the region deleted in the 17ΔN/H. Additional transcripts shown include the L/STs that end at the LAT polyadenylation site, as well as 0.7 kb LAT and UOL for which the polyadenylation sites could be the source of the 3SEQ10 hit.

Although the 2.0 kb LAT intron is not required for skin infection (**Fig 3F**), its splicing was confirmed during 17*syn*+ infection (**Fig 5C and S3 Table**). No 1.5 kb intron splicing was detected, in agreement with cell culture observations [12]. Notably, transcription abundance was higher at the 5′ end of the LAT locus (including exon 1) and 2 kb intron than at the 3′ end (including exon 2) (**Fig 5C and S3 Table**). RL2 was transcribed (**Fig 5 and S2 Table**) in both 17*syn*+ and 17ΔN/H infection. Although the 17ΔN/H deletion of the 5′ end of the LAT locus also removes the anti-sense 3′ end of the RL2 transcript (**Fig 1**), we saw that the splicing of RL2 was not affected by the deletion (**Fig 5D and S3 Table**). This was further assurance that ICP0 was functional during the infection, in addition to verifying that ICP0 retained its ubiquitin ligase function (**Figs 1 and 2C**). During both 17*syn*+ and 17ΔN/H infection of human skin, RL2 intron 1 was present at higher read coverage than the exons (**Fig 5C and 5D and S3 Table**). There was no gap in transcription between RL1 and RL2 in either virus (**Fig 5C and 5D**), which aligns with a recent report of an additional ORF found in between these genes [47].

Transcription preceding the LAT locus was detected during 17*syn*+ infection. It was inclusive of the pre-miR-H1 sequences and continued at a low level until the LAT locus start at 118007 nt (**Fig 1**). Some of these upstream transcripts likely continue into the LAT locus. However, there was a distinct difference in the magnitude of transcription preceding and at or after the LAT locus start site. These data signify that HSV-1 produces independent upstream LAT transcripts during infection in skin, and that these are less abundant than the 8.3 kb or a smaller LAT initiating at the LAT start site (**Fig 5C and S3 Table**).

17ΔN/H lacks the LAT promoter, exon 1, and a part of the 2 kb intron sequences, which are replaced by a 441 nt sequence from bacteriophage lambda [28] (**Fig 1 and S1 Table**). A small number of reads mapped to the lambda phage DNA insertion, with no strand preference. These were flanked by gaps in transcription indicating a lack of read-through (**Fig 5D and S2 Table**). Surprisingly, transcription at the remaining part of the LAT intron and exon 2 was not blocked despite the deletion of the 5′ end of the locus in 17ΔN/H (**Fig 5D, S2 and S3 Tables**). These data support the existence of additional elements driving transcription in the 3′ LAT end of the LAT locus [17,18]. Transcription was detected anti-sense to RS1 (ICP4) beyond the LAT polyadenylation site (**Fig 5C and 5D**), suggesting that the longer L/ST transcripts may also be present [17,18].

We also examined the possibility of LAT anti-sense transcripts other than RL1 and RL2 being expressed in skin. LAT anti-sense transcription, opposite to LAT exon 1 and the LAT intron, was detected during 17*syn*+ infection (**Fig 5C**). However, it was continuous and did not seem to be specific to the reported locations for AL-RNA, AL2, and AL3. These transcripts may be of extremely low abundance, if present at all (**Figs 1 and 5C**), as the coverage was markedly lower than for the UL56 5′ UTR (**Fig 5C**). This anti-sense transcription is likely

related to miR-H6 pri-miRNA, which is uncharacterised to date. This region is deleted in 17ΔN/H (**Fig 5C and 5D** and **S2 Table**).

Overall, our results indicated that all known HSV-1 genes, including LAT, are transcribed during skin infection with 17*syn*⁺. These data did not reveal differences in the transcriptomes of 17*syn*⁺ and 17ΔN/H beyond the LAT locus.

**Mapping of polyadenylated viral transcripts in 17*syn*⁺- and 17ΔN/H-infected skin.** As many transcripts overlap each other in the LAT locus, total RNA sequencing alone cannot be used to determine whether smaller LATs are present during infection. Moreover, the G+C-rich nature of the $TR_L$/$IR_L$ region in which LAT maps results in variable levels of sensitivity to detecting transcripts. Our approach to overcome these issues was to sequence the polyadenylated viral transcripts at their 3′ end using 3SEQ [48], which largely overcomes the effects of any biases due to transcript length, nucleotide composition, or secondary structure. Using 3SEQ, we were able to map low abundance RNAs and identify transcripts missing in 17ΔN/H-infected cells.

We confirmed the presence of transcripts ending at all annotated polyadenylated HSV-1 transcripts, including LAT, in both 17*syn*⁺ and 17ΔN/H (GenBank accession JN555585.1) (**Fig 5C and 5D** and **S4 Table**). Thus, we verified that transcripts ending at the LAT polyadenylation site are transcribed in the absence of the LAP1 promoter in 17ΔN/H (**Figs 5D and S2 and S4 Table**). These may include L/STs transcripts, coding for ORF-O and P, that can be polyadenylated at the LAT polyadenylation site [17,18]. Smaller, previously reported polyadenylated LAT transcripts designated AL-RNA, AL2, and AL3, and α0 intron 1 RNAs (**Fig 1**) were not detected.

We also found previously unannotated viral transcript ends in HSV-1 infected skin, which we designated as 3SEQ1 to 12 (**Fig 5C** and **S4 Table**). Transcripts ending at 3SEQ3, 3SEQ10, and 3SEQ12 were detected in another recent study [49]. We found that 3SEQ1 was of extremely low incidence and omitted it from further investigation (**Fig 5C** and **S4 Table**). Of interest, 3SEQ10 is found at a location matching the previously described 0.7 kb transcript [15] (**Figs 1 and 5C, and S4 Table**). Others have also detected this transcript in a transcriptome analysis of strains 17*syn*⁺ and KOS in cell culture [47,49,50]. We added a hypothetical transcript from the LAT start to 3SEQ10 into the Salmon analysis in order to determine whether there are transcripts overlapping LAT that terminate at 3SEQ10 (**S1 Fig**; listed as 5′ LAT). The quantification based on read coverage depth showed that these transcripts were expressed at greater levels than full-length or spliced LAT, and also supported the existence of a smaller transcript(s) in the 5′ end of the LAT locus. No 3SEQ region hits mapped between UL55 and LAT or on the opposing strand between RL2 and UL56. Independent confirmation of the proposed 3SEQ transcripts was not carried out, as all the viral transcript ends that we identified, except 3SEQ10 and SEQ11 located within the NotI-HpaI region deleted in the 17ΔN/H genome, were generated by both 17*syn*⁺ and 17ΔN/H (**Fig 5B and 5D**). No polyadenylated transcripts unique to 17ΔN/H were found.

In summary, we found that HSV-1 produces novel polyadenylated transcripts in skin, some of which overlap the LAT locus. All previously characterised polyadenylated transcripts are expressed in both 17*syn*⁺- and 17ΔN/H-infected skin. The only 17*syn*⁺ transcripts missing during 17ΔN/H infection were those located within the NotI-HpaI region deleted in the 17ΔN/H genome.

**Impaired pathogenesis of 17ΔN/H in skin is not associated with differential regulation of HSV-1 transcription.** Although we detected expression of all HSV-1 transcripts in skin infected with 17*syn*⁺ and 17ΔN/H using total RNA-seq, this did not constitute definitive evidence against LAT having effects on viral gene expression in skin. Differences in lesion formation during skin pathogenesis of wild-type and mutant viruses presents a multifaceted

challenge in quantifying and comparing the transcriptomes in a biologically relevant manner. Smaller lesions in the tissues infected with 17ΔN/H (**Fig 2**) result in lower amounts of viral RNA (**Fig 5**). For this, a simple normalisation to total transcripts is not a sufficient approach, as the number of newly infected and late infected or lysed cells depends on lesion size. This in turn may affect the proportion of transcripts from different kinetic classes (immediate-early, early, and late) present [51]. To address this, we performed laser capture microdissection (LCM) on equal-size skin lesions infected with 17ΔN/H, 17*syn*+, 17-37, and m2kb-S in order to represent best the same stage of infection for more accurate quantification of the transcriptomes using 3SEQ. This allowed viral transcripts to be amplified at equal ratios and the resulting data to be used for quantifying the transcripts. 17-37 and m2kb-S were included in the analysis because they had the same replication phenotype as 17*syn*+, and also to facilitate further investigation of whether LAT 2 kb intron splicing had any effect on viral gene expression.

All viruses had transcripts ending at the annotated 17*syn*+ viral gene polyadenylation sites (GenBank accession JN555585.1) (**S4** and **S5** **Tables**). Transcripts ending at the LAT polyadenylation site were detected in skin infected with each of the four viruses, including 17ΔN/H, which lacks the LAP1 promoter (**S5 Table**). These findings affirm the existence of independent promoters in the 3′ end of the LAT locus [17,18], in addition to the canonical LAT promoter (s) at the 5′ end. However, since transcription in the 3′ end of the LAT locus did not differ between 17*syn*+ and 17ΔN/H by RNA-seq or 3SEQ, this region is unlikely to have a virulence role in skin.

We hypothesised that in order for any significant up- or downregulation of viral transcripts to be the cause of the 17ΔN/H growth defect skin (**Fig 3A and 3B**), 17ΔN/H should have a significant difference in transcription of the particular gene compared to the three viruses, 17*syn*+, m2kb-S, and 17-37, which were not impaired for skin infection (**Fig 3**). In addition, that gene should not have a significant difference in expression between the three viruses that have a robust pathogenesis phenotype in skin: 17*syn*+, m2kb-S, and 17-37. We observed no such transcript.

We found that while transcripts ending at 3SEQ4 were significantly upregulated in 17ΔN/H compared to 17*syn*+-infected skin, there was no difference in its expression in 17ΔN/H compared to m2kb-S or 17-37 (**S5 Table**). These transcripts were also significantly upregulated in 17-37-infected compared to 17*syn*+-infected skin, and thus it is unlikely that they are regulated by LAT. Another variation in gene expression between the three viruses that were not impaired for skin infection was the downregulation of the UL44-UL45 transcripts in 17*syn*+- and 17-37- or m2kb-S-infected skin (**S5 Table**).

The UL33-UL34-UL35 transcripts were significantly upregulated in 17ΔN/H- compared to 17*syn*+-infected skin cells (**S5 Table**). However, the lack of difference in these transcripts between 17ΔN/H- and m2kb-S- or 17-37-infected skin suggests that the differential regulation between 17ΔN/H and 17*syn*+ is unlikely to be implicated in the 17ΔN/H pathogenesis defect. The UL56 transcript was significantly downregulated in 17ΔN/H- compared to m2kb-S-infected skin but not compared to 17*syn*+- or 17-37-infected skin (**S5 Table**) indicating that the observed variability was not biologically significant for pathogenesis. The UL31-UL32 transcripts were upregulated in 17ΔN/H- compared to 17*syn*+- and m2kb-S-infected skin (**S5 Table**). Despite this, these transcripts were expressed at the same levels in 17ΔN/H- and 17-37-infected skin. Overall, no gene was differentially expressed in 17ΔN/H- and 17-37-infected skin (**S5 Table**). Therefore, although significant differences were detected in the levels of expression of genes between 17ΔN/H and 17*syn*+ or m2kb-S in human skin, the lack of differences between 17ΔN/H and 17-37 suggested that these are not the primary cause of the defect in 17ΔN/H pathogenesis.

Moreover, the regulation of gene expression did not appear to be due the 2 kb LAT intron, since the gene expression of 17ΔN/H and m2kb-S did not show a similar pattern for UL31-UL32 and UL56 expression, which would have been expected given that 17ΔN/H lacks a half of the 2 kb intron and m2kb-S is deficient in its splicing. Thus, we demonstrated that neither deleting the start of the LAT locus nor mutating the 2 kb LAT intron splice acceptor site to abrogate splicing had effects on the HSV-1 transcriptome in skin comparable to those reported in neurones [31–33].

In summary, no HSV-1 gene was significantly up- or downregulated in ΔN/H infected-skin compared to the three viruses that exhibited wild-type pathogenesis in skin.

## Discussion

The LAT locus is abundantly transcribed in neurones and produces a 2 kb intron known as the hallmark of HSV-1 latency. The locus has been implicated in inhibition of apoptosis, regulation of viral gene expression, and reactivation from latency in mouse or rabbit neuronal models *in vivo*. Despite its expression during lytic infection in cell culture, it has not been studied in the context of lytic infection in human epithelial tissues. Here, we studied HSV-1 pathogenesis and the LAT locus in human skin using an *in vivo* skin xenograft model, supplemented with an explant culture model. We used three different LAT mutant viruses, 17ΔN/H, Δ1773, and KOSΔPstLAT, with deletions in the 5′ end of the LAT locus, and compared them to viruses with an intact LAT locus. We discovered that the LAT locus has a previously unrecognised role in the pathogenesis of HSV-1 in human skin. This was evident from the dramatically impaired lesion formation and replication exhibited by the viruses lacking the 5′ end of the LAT locus during lytic infection of human skin tissue *in vivo*.

The transcriptional complexity of the LAT locus has made defining the mechanisms by which the LATs contribute to HSV-1 pathogenesis very challenging in neurones. Moreover, LATs may be expressed in a cell-type specific manner, as has been shown for LAT miRNAs that regulate viral transcripts [19–21]. In contrast to previous work conducted in mouse neurones where LAT has been shown to be a factor in regulation of viral gene expression [31–33], we found only three differences in the levels of known transcripts resulting from a deletion of the 5′ start of the LAT locus (increased UL31-UL32 and UL33-UL34-UL35) or inhibition of LAT 2 kb intron splicing (decreased UL44-UL45) compared to 17*syn*$^{+}$ in skin. Moreover, these were not related to the pathogenesis phenotypes of the mutant viruses, based on our differential gene expression analysis. The role of LATs in gene regulation can be expected to differ in epithelial cells and neurones due to the nature of HSV-1 gene expression and silencing. For example, the key transactivating protein VP16 is present in the tegument and can function immediately upon viral entry in primary infection, whereas it must be synthesised *de novo* during reactivation [52]. Hence, the ability of LATs to control gene expression would be redundant in a primary infection.

Our finding of the importance of the LAT locus in human skin infection is novel because LAT mutants have not been shown to be defective in replication at peripheral sites in mouse models [30,33]. A virus with a deletion of the 5′ end of LAT in strain KOS was shown to have viral titres comparable to its parental and rescuant virus in eye swabs from acutely infected CD-1 mice [30]. Also, deletion of the LAT promoter does not appear to affect replication of a strain SC16 virus engineered to express firefly luciferase in whisker pads [33], but this may not be unexpected as LATs are not detected in the epidermis of footpads [13,53]. It may be possible that there is a species-specific element to the importance of the locus. The human epidermal architecture and gene regulatory networks differ substantially from those of mouse skin [54]. The human epidermis is thicker, cell turnover slower, and hair follicles less dense. Consistent

with clinical observations, lesion formation requires seven days in human skin xenografts [43] but is accelerated in mouse models. Species-specificity has previously been shown for some key HSV-1 interactions with host cells, such as necroptosis and antigen presentation [55,56]. Regardless, some functions of the LAT locus may be shared across species and cell types. The LAT locus has been associated with inhibition of apoptosis in mouse models [27,34], and the LAT deletion mutant 17ΔN/H exhibited increased levels of apoptosis in human dendritic cells [57]. Although the difference was subtle, release of virus from keratinocytes was lower for 17ΔN/H than 17*syn*+, as has been observed previously in mouse trigeminal ganglia explants undergoing reactivation [28]. Diminished release of HSV-1 has the potential to impact viral spread to neighbouring epithelial cells within skin as well as in ganglia, since extracellular virions were detected in dorsal root ganglia infected with HSV-1 [58]. A similar scenario is possible during lytic infection in human skin, where a decrease in virus release would impact the lesion formation.

Our analysis of the HSV-1 transcriptome in human skin tissue *in vivo* complements other recent studies that have improved our knowledge of HSV-1 transcription in HFs [47,49–51]. We demonstrated a loss of virulence in human skin for mutants lacking the 5′ end of the LAT locus, noting that transcription in the 3′ end of the LAT gene was unchanged. Hence, the cause may be narrowed down to an element in the 5′ end. The absence of one or more transcripts originating from the 5′ end of the LAT locus, such as the transcript(s) we detected ending at the polyadenylation site 3SEQ10, could be involved in the defective pathogenesis. No transcript(s) ending at 3SEQ11 site has been described, but any such would be unlikely to be relevant due to the low expression (**S4 Table**). A previously described transcript (0.7 kb LAT) that matches 3SEQ10 in location [15], and that has also been detected in HSV-1-infected HFs [47,49,50], is of particular interest because it appears to contribute to virulence in a mouse corneal inoculation model [15]. A caveat is that the miR-H1/H6 deletion did not affect pathogenesis in skin, despite removing 25 nt within the 0.7 kb LAT RNA [36]. Although a role for miR-H15 cannot be excluded [19], its absence is unlikely to be the sole cause of the marked growth defect of LAT mutants with deletions in the 5′ end of the LAT locus, as miRNAs are generally known to cause small modulatory effects. It is noteworthy that these deletion mutants also remove the binding sites for transcription factors CREB, CTCF, and SP1 from the 5′ end of the LAT locus [59]. It is not immediately clear what effect this may have had, since the HSV-1 transcriptome, including the 3′ end of the LAT locus, was not affected by the deletion of the 5′ end of the LAT locus. Nevertheless, their involvement remains a possibility.

Intriguingly, ribosomal subunits associate with sequences at the 5′ end of [47] and at the 2 kb intron [60], indicating potential protein synthesis, and proteins encoded within the LAT locus have been reported but not fully characterised [61–64]. We detected transcripts ending at previously uncharacterised polyadenylation sites in the proximity of the LAT locus in human skin by 3SEQ (3SEQ9, 3SEQ10, and 3SEQ11), which may also express proteins. Together, these findings warrant renewed investigation into protein expression from the LAT locus. It would be of great interest to compare the HSV-1 transcriptome and proteome during lytic and latent infection in differentiated human skin and neuronal tissue. Information on the precise identities and start and end sites of LATs would guide the design of relevant mutants in order to disrupt them selectively. Refined LAT locus mutants will be necessary to probe LAT-related effects on host cell proteins in cell culture models and to assess further the consequences for HSV-1 pathogenesis *in vivo*.

A fundamental finding from our study is that viral genome sequencing is important for future studies of HSV-1 pathogenesis and of viral pathogenesis in general. This approach revealed many unsuspected genetic changes—alterations in proteins of all kinetic classes and various functions—that did not diminish the capacity of HSV-1 to infect human skin *in vivo*.

Notably, the 17-37 and m2kb-S viruses derived from the 17-37 BAC had a mutation leading to a truncation of gC. This observation aligns with the recent report of multiple gC variants in the original Glasgow 17*syn*⁺ strain stock and in many derived virus stocks used in different laboratories [38]. Despite the gC mutation in the 17-37 and m2kb-S viruses, their replication in skin was not impaired, whereas our early work had suggested a role for gC in skin [65]. We hypothesise that the previous HSV-1 gC mutant, which was extensively passaged but not sequenced, had other mutations that diminished skin pathogenesis. HSV-1 genomes accumulate genetic changes that become apparent with serial passage, for example SNPs that lead to increase in syncytia formation [66]. Using the HSV-1 BAC for generating new mutant viruses is a practical method that minimises undesired genetic changes, although sequencing of the derived viruses remains important because some mutations may occur even at low passage numbers, as we noticed with the 17-37 and m2kb-S viruses. Generation of mutant and rescuant viruses, by whatever means, should be accompanied by genome sequencing, although generation of rescuants is unnecessary for any mutant that is sequenced alongside the parental virus.

In conclusion, we found that the LAT locus is a virulence determinant for HSV-1 infection in differentiated human skin *in vivo*. Overall, our study reveals that deletions in the locus result in abortive or delayed viral spread during primary infection. Information on the functions of the LAT locus could be used for new antiviral strategies to limit the formation of primary and recurrent lesions. These findings are relevant for the design of an attenuated HSV-1 vaccine candidate, as disrupting the skin pathogenesis could lead to fewer neurones harbouring latent viral genomes. In the absence of LAT, latent genomes would be predicted to have a reduced capacity for reactivation, and should reactivation occur, replication and lesion formation would be severely impaired.

## Methods

### Ethics statement

NIH guidelines for housing and care of laboratory animals were followed (Animal Welfare Assurance # A3213-01), Institutional Animal Care and Use Committee (IACUC) review of research involving animals was performed, and procedures were approved by the Stanford University Administrative Panel on Laboratory Animal Care (Protocol ID # 11130). Human foetal skin tissue was obtained from Advanced Bioscience Resources with informed consent in accordance with US federal and state regulations. Acquisition and use of foetal material have been reviewed by the Stanford University Administrative Panel on Human Subjects in Research and do not meet the criteria for research involving human subjects. The Institutional Review Board and Institutional Biosafety committees at SUNY Upstate Medical University reviewed and approved the use of human foetal tissue for this project, which did not meet the criteria for human subjects' research.

**Viruses.** All viruses derived from HSV-1 strain 17*syn*⁺, including our 17*syn*⁺ stock (a gift from Roger Everett), were sequenced, and their genomes were compared to the reference sequence for this strain (GenBank accession JN555585.1).

17ΔN/H. 17ΔN/H (**Fig 1**; a gift from Nigel Fraser) has a 2026 nt NotI-HpaI deletion in the 5′ end of the LAT locus (5902:7927 and 118447:120472) replaced by a 441 nt sequence from lambda phage DNA [28]. Although a repaired virus had also been made, it was no longer in the possession of the first or corresponding authors [28].

17ΔPst and 17ΔPstR. 17ΔPst was supposed to have a 203 nt deletion of a PstI-PstI fragment within the LAT promoter LAP1 (**Fig 1**; a gift from David Bloom) [35]. However, our genome sequencing showed that it contained a larger 1773 nt deletion in the 5′ end of the LAT locus

(7116:8888 and 117486:119258). Therefore, we renamed the virus Δ1773. 17ΔPstR was reported to be the rescuant for 17ΔPst [35]. However, our genome sequencing showed that it was not derived from Δ1773.

17dmiR-H1/H6 and 17REdmiR-H1/H6. 17dmiR-H1/H6 has a 25 nt (GGGGATGGAAGG ACGGGAAGTGGAA) deletion spanning 8022:8046 and 118328:118352 (**Fig 1**; a gift from David Bloom) [36]. This deletion removes the seed sequences of the two anti-sense miRNAs miRH1/H6: miR-H1-5p (miRbase MIMAT0003744; 8023:8043 and 118331:118351) and miR-H6-3p (MIMAT0008404; 8025:8045 and 118329:118349) [19, 20]. The 17REdmiR-H1/H6 virus was generated in the Bloom lab to rescue the mutation in 17dmiR-H1/H6 and provided to us ahead of publication. The methods used to generate 17REdmiR-H1/H6 were those described for 17dmiR-H1/H6 [36]. Primers BS-H1-5f TCCCCCGGGCTGCAGGAATTCCC AGTCTCCTCGCCTTCTC and BS-H1-3r GATAAGCTTGATATCGAATTCCTGCCTCTG CCGCTTGTG were used to construct a recombination plasmid with an intact miR-H1/H6 sequence that was then co-transfected with 17dmiR-H1/H6 virion DNA to rescue the deletion. Although 17REdmiR-H1/H6 was provided as the rescuant from 17dmiR-H1/H6, our genome sequencing showed it was not derived directly from 17dmiR-H1/H6.

KOSΔPstLAT and KOSΔPstR, derived from strain KOS rather than strain 17$syn^+$. KOSΔPstLAT has a deletion of a PstI-PstI fragment within the LAT promoter, and KOSΔPstR (**Fig 1**; a gift from David Knipe) [45] is the rescuant. We verified the deletion and its rescue using PCR and sequencing, which showed that the KOSΔPstLAT deletion is 203 nt, leaving a single PstI site in the LAP1 promoter (7481:7683 and 118558:118760 in GenBank accession JQ673480.1). Genome sequencing has not been performed by us or others.

**Creation of m2kb-S virus with a mutation of the LAT splice acceptor site.** Two selectable markers were introduced into the HSV-1 17-37 BAC (a gift from David Leib) [37] by red recombination in order to mutate the LAT 2 kb intron splice acceptor site at both genome locations [67]. Three substitutions were introduced at the LAT 2 kb intron splice acceptor site at 121420, 121421, and 121424 nt in IR$_L$, and the corresponding positions in the TR$_L$. These mutations have been shown previously to abrogate splicing of the intron in lytic infection in another mutant [68]. Primers were designed to maintain the RL2 coding region antisense to the LAT splice site, as previously described [68]. Two primers were generated to mutate one copy of the splice site using Kan as a selectable marker: Mut LAT SA F (kan) 5′ CGGGGG CCGAGGGAGGTTTCCTCTTGTCTCCCTCC CA*AA*G*C*T*CC GACGGCCCCGCCCGAG AGGATGACGACGATAAGTAGGGATA 3′ and Mut LAT SA R (kan) 5′ TCCTCCTCCG CTTCCGCCTCCTCGGGCGGGGCCGTCGG*A*GC*TT*TGGG AGGGAGACAAGAGCAAC CAATTAACCAATTCTGATTAG 3′. Bold, italic nucleotides indicate changes to maintain the RL2 (ICP0) coding region, mutation of the splice acceptor site, and introduction of an AluI restriction site. Underlined nucleotides are the start and end of the Kan cassette. PCR was done using pEP-KanS2 [67] as the template and products were cloned into the pCR-4 TOPO cloning vector (Thermo Fisher Scientific). Mutations were verified by sequencing, the DNA cassette was amplified by PCR, gel purified and electroporated into GS1783 cells (Thermo Fisher Scientific) containing the 17-37 BAC and selected in the presence of chloramphenicol and kanamycin (1st red recombination). Colonies were screened by PCR using the Platinum SuperFi PCR Master Mix (Thermo Fisher Scientific cat. # 12368050). GS1783 cells were electroporated with a clone positive for the Kan cassette insertion and competent cells were prepared. A second set of primers, similar to those above, was designed to amplify the zeocin cassette. PCR and TOPO cloning were done using the pGIPZ vector (Thermo Fisher Scientific) as a template, DNA was electroporated into GS1783 cells containing the 17-37 BAC with the Kan insertion and selection was done in the presence of chloramphenicol, kanamycin, and zeocin. Clones containing both the Kan and zeocin resistance markers were used for the 2nd

red recombination step to excise the selection markers and introduce the splice site mutation at both locations. Mutations were verified by sequencing and by the presence of the AluI site. Positive BAC clones were grown and purified using the Qiagen Large Construct Kit. Vero-CRE cells were transfected with BAC DNA using Lipofectamine 2000 (Thermo Fisher Scientific cat. # 11668019) to excise the BAC sequence and generate 17-37 and m2kb-S virus stocks that were subsequently plaque purified. Both genomes were sequenced.

**Cell culture assays.** Viruses were propagated and titered in Vero African green monkey kidney epithelial cells obtained from ATCC (American Type Culture Collection) that were cultured in DMEM supplemented with 10% foetal bovine serum (heat inactivated) and penicillin/ streptomycin and amphotericin (Thermo Fisher Scientific). The plaques were evaluated by immunohistochemistry using anti-HSV antibody (Accurate Chemical and Scientific Corporation cat. # AXL237/2 lot # J21880). The cell monolayers were inoculated with 10-fold serial dilutions of the viruses for 1 h at 37˚C, followed by PBS wash and overlaid with media containing 1.5% carboxymethyl cellulose. Replication and virus release assays were done using human foreskin fibroblasts (HFs; obtained from ATCC) or primary keratinocytes isolated from foreskin (a gift from Paul Khavari). HFs were cultured in the same media as Vero cells. Keratinocytes were cultured in keratinocyte-SFM mixed 1:1 with media 154 and supplemented with human keratinocyte growth supplement (HKGS) and amphotericin B (Fungizone) according to the manufacturer's instructions (Thermo Fisher Scientific cat. # 17005042, M154500, S0015, and 15290026). In the replication assays, cell monolayers were inoculated with viruses for 1 h at 37˚C (MOI 1), followed by PBS wash and overlay with the culture media. At harvest (3, 6, 12, or 24 HPI), cells were scraped into the medium and exposed to two freeze-thaw cycles prior to titration. Virus release assays were inoculated similarly for 1 h at 37˚C (MOI 0.0001) and the supernatant was harvested at 24, 48, and 72 HPI, followed by supernatant titration without freeze-thaw cycles.

## 2 kb LAT intron splice assay

Vero cells in a T75 flask (approximately 3 million cells) were infected with 17$syn^+$, 17-37, or m2kb-S at MOI 1. An uninfected control flask was included as a control. Total RNA was extracted at 24 HPI using Trizol (Thermo Fisher Scientific cat. # 15596026), and treated with DNase I, RNase-free (Thermo Fisher Scientific cat. # EN0521), according to the manufacturer's instructions. First-strand synthesis was performed using a gene-specific primer with or without SuperScript III reverse transcriptase (Thermo Fisher Scientific cat. # 18080–093), followed by PCR using KOD Xtreme Hot start DNA polymerase (EMD Millipore cat. # 71975–3) according to the manufacturer's instructions. A water-only control was included in PCR reactions.

The level of unspliced 2 kb LAT intron during 17$syn^+$, 17-37, or m2kb-S infection was detected using RT-PCR. First strand synthesis was performed using a gene-specific reverse primer 5′ TCTTCCTCCTCTGCCTCTTCC 3′ that binds within LAT 3′ exon 2 (121514: 121534 in GenBank accession JN555585.1). Subsequent PCR was done using the same reverse primer combined with a forward primer 5′ CGGGTACTCGGGGGGGCA 3′ that binds within the 2 kb intron (121049:121065). This resulted in a 486 bp product across the 2kb LAT intron splice acceptor site when the transcript is not spliced. The absence of a product indicated efficient splicing.

As a control, we performed a similar analysis using primers specific for RS1 (ICP4) and UL54 (ICP27). For RS1, first strand synthesis was done using a forward primer 5′ GGCGG GAAGTTGTGGACTGG 3′ that binds 127292:127311 in IR$_L$. The same primer was paired with a reverse primer 5′ CAGGTTGTTGCCGTTTATTGCG 3′ in the PCR reaction, which

binds 127174:127195 in IR$_L$. The product size is 138 bp. For UL54, first strand synthesis was done using a forward primer 5′ TTTCTCCAGTGCTACCTGAAGG 3′ that binds 114923: 114944. The same primer was paired with a reverse primer 5′ TCAACTCGCAGACACGAC TCG 3′ in the PCR reaction, which binds 115185:115205. The product size was 283 bp. The markers used to assess the PCR product sizes were Gene ruler 1 kb plus (Thermo Fisher Scientific cat. # SM1333) and 100 bp (cat. # SM0323).

**Western blotting.** Vero cells were inoculated with 17*syn*⁺ or 17-37 (MOI 1). At 24 HPI, cells were washed with PBS and lysed with 3x lysis buffer (30% stacking gel buffer [0.5 M Tris with 0.4% SDS, pH 6.8], 30% glycerol, 6.5% SDS powder, 9M urea, 100mm DTT and bromophenol blue) diluted to 1x with PBS. Samples were run on a 4–20% mini-PROTEAN TGX precast protein gel (Biorad, cat. # 4561093) and transferred onto an Amersham Protran nitrocellulose membrane (Sigma-Aldrich, cat. # GE10600002). Blocking was done in 5% nonfat milk. The gC and β-actin (control) proteins were detected using HSV-1 anti gC antibody clone M701139 (Fitzgerald, cat. # 10-H25A, batch 912, mouse monoclonal, 1:1000 dilution), anti-β-actin antibody clone AC-15 (Sigma, cat. # A5441, lot no. 127M866V, mouse monoclonal, 1:1000 dilution), and anti-mouse IgG, human ads-HRP antibody (Southern Biotech, cat. # 1030–05, lot no. K3515-T566G, goat, 1:1000 dilution).

**HSV-1 genome sequencing.** 500 ng viral genomic DNA extracted from purified nucleocapsids [69] was sonicated in a Covaris Model S2 sonicator (intensity 5, duty cycle 10, cycle per burst 200, treatment time 180 s). Sequencing libraries were made using SPRIworks System I for Illumina Genome Analyzer (Beckman Coulter cat. # A88267) and Illumina DNA barcodes. The libraries were pooled at equal concentrations and sequenced on an Illumina MiSeq with 2x75 nt reads. Sequencing adapters and low-quality reads were removed using Trim Galore v. 0.4.0 (https://github.com/FelixKrueger/TrimGalore). Bowtie 2 v. 2.3.1 [70] was used to align the reads against the HSV-1 strain 17*syn*⁺ genome sequence (GenBank accession JN555585.1) trimmed of all but the 100 nt of TR$_L$ and TR$_S$ adjacent to U$_L$ and U$_S$, respectively, in order to avoid ambiguities in assembly. The assemblies were reviewed using Tablet v. 1.19.09.03 [71]. Sequence data were typically not obtained across large tandem reiterations. Proportions of differences were scored only for differences that affect protein-coding either by substitutions or by frameshifts (indels) by counting reads containing specific sequences or their complements. The data can be accessed in NLM BioProjects: PRJNA338711 and PRJNA544722 (17dmiR-H1/H6 only).

**Skin xenografts.** Human foetal skin was cut into 1 cm² pieces, which were implanted subcutaneously in male *scid/scid* (SCID) homozygous C.B-17 male mice aged 6–8 weeks (Taconic Biosciences) [72], inoculated with virus by scarification at 5 weeks post-engraftment, and processed for viral titres and formalin-fixed, paraffin-embedded (FFPE) tissue blocks upon harvest. Mice were anaesthetised by intraperitoneal injection with a solution of 5% (w/v) ketamine and 2.5% (w/v) xylazine.

**Tissue section staining, *in situ* hybridisation, and imaging.** Alternate 6 μm tissue sections were stained with haematoxylin and eosin (H&E), HSV DNA BIO-PROBE (Enzo cat. # ENZ-40838) or LAT ATD19 RNA probe [73] by *in situ* hybridisation, and by antibodies. Antigen retrieval using a citrate buffer in a pressure cooker was performed prior to staining sections with antibodies to the viral ICP0 and cellular PML proteins (Santa Cruz HSV-1 ICP0 Antibody (11060) mouse monoclonal IgG2b cat. # sc-53070 lot # A0313 and PML Antibody (H-238) rabbit polyclonal IgG cat. # sc-5621 lot # J2010). H&E staining and *in situ* hybridisation images were captured using a Leica DM2000 with PL FLUOTAR 10X/0.3 (cat. # 506505) lens and camera DFC290HD. Software used for image acquisition was Leica Application Suite 4.3.0 (Leica Microsystems CMS GmbH). Antibody stain images were captured on a Leica SP5 confocal microscope with an HCX PL APO CS 40.0x 1.25 OIL UV lens and a Leica Hybrid

detector in Stanford Cell Sciences Imaging Facility. Software used for image acquisition was Leica LAS AF 2.7.3.9723 (Leica Microsystems CMS GmbH).

**Skin explants grown in tissue culture.**   Human foetal skin was cut to 1 cm² pieces that were scratch-inoculated with a virus on a petri dish and incubated for 3 h at 37˚C prior to lifting onto NetWell inserts (Corning cat. # 29442). Tissues were incubated at the liquid–air interphase with the epidermis exposed to air, replacing the media every 2 days, at 37˚C in humidified 5% $CO_2$ until harvesting at 4 or 8 DPI [74]. The culture medium was DMEM containing 10% foetal bovine serum (heat inactivated) and penicillin/streptomycin and amphotericin.

**RNA-seq.**   Total RNA was extracted using TRIzol from 17*syn*⁺- and 17ΔN/H-infected (2,500 PFU) skin xenografts at 11 DPI. Libraries were prepared from 1 µg of each RNA sample using the KAPA Stranded RNA-seq with RiboErase kit (Roche cat. # KK8483), pooled at equal nanomolar concentrations, and sequenced on an Illumina NextSeq 500 Mid Output with paired-end 75 nt reads. Read pairs from the 17*syn*⁺-infected skin were aligned to a reference sequence combining the human hg38 and 17*syn*⁺ (GenBank accession JN555585.1) genomes, with human SNP annotations from dbSNP release 146 and transcript annotations from GENCODE release 25. Read pairs from the 17ΔN/H-infected skin were aligned to a reference sequence combining the human genome and a modified viral genome with a 2026 nt NotI-HpaI deletion in the 5′ end of the LAT locus (5902:7927 and 118447:120472) replaced by a 441 nt sequence from lambda phage DNA [28]. The 17ΔN/H genome annotations were made based on the modified 17*syn*⁺ reference genome. Alignments were done using STAR release 2.5.2a using ENCODE settings and excluding multiple alignments [75]. Duplicate read pairs were removed using the rmdup function of SAMtools 1.3.1 [2] [76]. Strand-specific coverage plots were generated using BEDTools 2.25.0 [3] [77] after assigning each read pair to its strand of origin according to the orientation of the read alignments: those with read 1 on the − strand of the reference genome and read 2 on the + strand were assigned to the + strand, and vice versa, consistent with the orientation of the strand-specific library synthesis. Read pairs with discordant alignments were excluded. Gene expression was quantified by Salmon 0.8.2 [46], using a custom reference sequence in which both human and viral genomes were included to allow removal of ambiguous reads. Salmon parameters were set to consider read orientation and correct for biases due to G+C content and random primer sequences. Viral transcripts were extracted from the transcripts per million (TPM) values calculated by Salmon. Host genes were excluded from further analyses to eliminate confounding by the variable amount of total viral RNA captured in each dissection. Thus, the viral transcripts were re-normalised to the total viral transcripts; and the abundance for each transcript was reported as transcripts per thousand (TPT) due to the small number of viral compared to host genes. The data can be accessed in NLM BioProject: PRJNA338711.

In addition, splice events in the LAT and RL2 (ICP0 protein) transcripts in the 17*syn*⁺- and 17ΔN/H-infected skin xenografts were quantified using the start and end coordinates of all read pairs in the viruses' RNA-seq datasets.

**Laser capture microdissection (LCM).**   FFPE skin tissue blocks were cut on a microtome, placing 7 µm sections onto polyethylene naphthalate membrane glass slides (Thermo Fisher Scientific cat. # LCM0522). The sections were stained using haematoxylin. Viral DNA and RNA were detected in adjacent 7 µm sections by *in situ* hybridisation (as described above) to determine where the lesions were located in each tissue sample. These data were then used as a guide to select an area of 300,000 µm² infected cells in the next adjacent 7 µm section for Arcturus XT laser capture. The cells were captured to a CapSure HS LCM cap (Thermo Fisher Scientific cat. # LCM0215). Caps were stored at -20˚C overnight. Arcturus XT was equipped with Nikon Eclipse Ti-E and the software used for acquisition was Arcturus XT Software v3.4.

**Smart-3SEQ.** Smart-3′-end sequencing for expression quantification (3SEQ) libraries were prepared according to the protocol for FFPE tissue on Arcturus LCM caps [48]. Briefly, captured tissue was lysed in place on the cap by lysis buffer incubation for 1 h at 60°C, and then the entire lysate was used for library preparation. No shearing of RNA was needed, as FFPE RNA is already fragmented. Oligo(dT) primer was used for first strand synthesis. Second strand cDNA synthesis was done via template switching and Illumina sequencing adapters were incorporated.

The cDNA was then amplified using PCR primers that added multiplexing barcodes. The libraries were cleaned using magnetic beads, and their quality was assessed using an Agilent 2200 TapeStation with High Sensitivity D1000 reagent kits. Only libraries with substantial yields above 200 bp were chosen for sequencing on an Illumina NextSeq 500 to generate 76 nt single-end reads. Reads were trimmed and aligned as described previously [48], using STAR 2.3.5a [75] and the combined human and viral references used for RNA-seq. Strand-specific coverage plots were generated by filtering reads by strand and computing depth per position with SAMtools 1.4 [76]. The data can be accessed in NLM BioProject: PRJNA338711.

**Counting reads in enriched regions.** The 3′ ends of viral transcripts were determined using UniPeak [78] that detects regions of high read density in alignments on a reference sequence. The 3SEQ reads were first aligned onto the human hg38 and the 17syn⁺ (GenBank accession JN555585.1) reference sequences. A modified 17syn⁺ reference sequence with a 2026 nt NotI-HpaI deletion in the 5′ end of the LAT locus (5902:7927 and 118447:120472) replaced by a 441 nt sequence from lambda phage DNA [28] was used for 17ΔN/H. Strand-specific regions of high read density were then detected in the unified coordinate system using pooled data from all infected libraries. Mock infections and bystander (uninfected cells near a lesion in the same tissue) samples did not contribute to the region calling; however, reads were still counted in those samples at the regions detected from the infected samples. The Unipeak parameters were set for a narrow bandwidth of 25 bp and sensitive detection threshold of 2-fold enrichment relative to uniform distribution of all reads over the common genome, relative to the default settings. In addition to the automatically detected regions, the entire lambda phage DNA insertion in the 17ΔN/H genome [28] was manually added to the list of regions to count any possible hits on either strand. Viral TPT (described in the RNA-seq methods) read counts per region were determined for the differential expression analysis performed using DESeq2 1.26.0 [79]. The DESeq2 parameters were set for an additive model with virus genotype and tissue donor as predictors, to control for variation among tissue donors. All samples, including bystander and mock infection, were used to fit the models in DESeq2; however, only the relevant xenograft samples were used in each two-class test of differential gene expression.

The numerical data used in all figures are included in S1 Data except for the genome sequencing, which can be found in S1 Table.

## Supporting information

**S1 Table. Amino acid and other sequence changes in 17syn⁺-derived viruses.** The genomes of 17syn⁺, 17ΔN/H, 17-37, m2kb-S, Δ1773, 17ΔPstR, 17dmiR-H1/H6 (dH1/H6), and 17REdmiR-H1/H6 (RE) used in the study were sequenced. Amino acid sequence changes found in each virus, the types of differences (SNP leading to substitution or indel leading to frameshift) and their locations in the 17syn⁺ genome (GenBank accession JN555585.1), are specified in the first spreadsheet. Substitution occurrence (%) is given where the lighter shading reflects low occurrence and the darker shading high occurrence. All SNPs and indels in non-coding and coding regions are listed in the spreadsheets following an overview of the datasets. Their locations in the 17syn⁺ genome are indicated (GenBank accession

JN555585.1 genome location column). In addition, the locations in the reference genome used in the alignment, which is trimmed of all but the 100 nt of $TR_L$ and $TR_S$ adjacent to $U_L$ and $U_S$, respectively, are given (reference location column). The changes are classified as minor, major, or complete, depending on the proportion of genomes affected.
(XLSX)

**S2 Table. Salmon analysis of viral transcripts from RNA-seq on 17*syn*[+]- and 17ΔN/H-infected skin.** Viral gene expression in infected human skin was analysed by RNA-seq and quantified by Salmon 0.8.2. Viral transcript abundances are reported as transcripts per thousand (TPT; described in the RNA-seq methods). Some viral transcripts that overlap other viral transcripts have zero values for either or both 17*syn*[+] and 17ΔN/H because their abundances relative to the highest coverage viral transcripts were low. Transcripts US1 and US12 share the exon 1 sequence, which therefore was not counted. LAT and RL2 transcript abundances were evaluated for both the spliced and unspliced species. In addition, three regions were added manually to identify any transcripts from these regions. These were the 5′ end of the LAT locus (5′ LAT) where the 3SEQ10 region hit is located, the 3′ end of the LAT locus (3′ LAT) that was not deleted in 17ΔN/H (118007:118735 in GenBank accession JN555585.1), and the lambda phage DNA insert in the 17ΔN/H genome.
(XLSX)

**S3 Table. Splicing of the LAT and RL2 (ICP0 protein) transcripts in 17*syn*[+]- and 17ΔN/H-infected skin.** Splices in the LAT and RL2 (ICP0 protein) transcripts in 17*syn*[+]- and 17ΔN/H-infected skin xenografts was investigated by counting cDNA fragments within the LAT (17*syn*[+] infection) or RL2 (17*syn*[+] and 17ΔN/H infection) loci. LAT has one intron, whereas RL2 has two introns. Each row is the feature where the cDNA fragment started, and each column is where the fragment ended. cDNA fragments that cover an intron to an exon junction, or vice versa, represent unspliced transcripts e.g. 163 cDNA fragments span from LAT exon 1 to intron 1. cDNA fragments that cover an exon to an exon junction without the intron sequence in between represent spliced transcripts e.g. 83 cDNA fragments spanned from LAT exon 1 to exon 2. Read pairs that were completely contained within an exon or an intron were also counted to compare abundance e.g. 4083 cDNA fragments were completely within the LAT intron because they began in the LAT intron 1 and ended in intron 1 on the diagonal axis.
(XLSX)

**S4 Table. Viral transcript counts from 3SEQ in 17*syn*[+]-, 17ΔN/H-, m2kb-S-, and 17-37-infected skin *in vivo* and cultured skin explants.** 3SEQ was performed on 17*syn*[+]-, 17ΔN/H-, m2kb-S-, and 17-37-infected xenografts and cultured skin explants (*ex vivo*). Viral transcript abundances are reported as transcripts per thousand (TPT; described in the RNA-seq methods). The lambda phage insertion in the 17ΔN/H genome was manually added to the analysis to identify any unknown polyadenylated transcripts arising due to the insertion, but none were found. The sample data were deposited in NLM BioProject: PRJNA338711 and the SRA repository FASTQ file name for each sample is given. Codes E1-3 for explants and G5, 6, 7, and 11 for xenografts were used to indicate individual skin donors.
(XLSX)

**S5 Table. Differential gene expression in 17*syn*[+]-, 17ΔN/H-, m2kb-S-, and 17-37-infected skin *in vivo*.** Differential gene expression between any two viruses out of 17*syn*[+] (n = 4), 17ΔN/H (n = 3), 17-37 (n = 3), and m2kb-S (n = 3) in infected skin xenografts was calculated using DESeq2 on 3SEQ data. For any gene, a positive $\log_2$-fold change indicates an increase in the first virus over the other, whereas a negative value indicates a decrease. An adjusted p value

(padj) of <0.05 indicates a significant result (yellow colour). 3SEQ10 and 3SEQ11 are not included in the comparisons because they are located within the region that is deleted in the 17ΔN/H genome. Raw counts are given for all samples, including those not used for differential gene expression analysis: the infected explants (*ex vivo*), mock infections, and bystanders. Codes E1-3 for explants and G5, 6, 7, and 11 for xenografts were used to indicate individual skin donors.
(XLSX)

**S1 Fig. Splicing of the 2 kb LAT intron during 17*syn*[+], 17-37, and m2kb-S infection.** Splicing of 2 kb LAT intron during 17*syn*[+], 17-37, and m2kb-S infection in Vero cells at 24 HPI (MOI 1). RT-PCR using gene-specific primers with or without reverse transcriptase (RT) was performed to verify lack of 2 kb LAT intron splicing in the splice mutant virus m2kb-S. Uninfected cells and water only were used as negative controls for all viral transcripts. RS1 and UL54 RNAs were detected as positive controls for viral infection.
(TIF)

**S2 Fig. Replication of Δ1773 and KOSΔPstLAT in cell culture.** Viral titres ($\log_{10}$ PFU/ml ± SEM) from replication assays in HFs at specified time points 3, 6, 12, and 24 HPI (MOI 0.5; n = 3 for both viruses per time point). 2-way ANOVA was performed with Sidak's correction, where all time points had non-significant P > 0.05 differences. (A) Δ1773 and 17ΔPstR and (B) KOSΔPstLAT and rescue KOSΔPstR virus comparison.
(TIF)

**S3 Fig. Replication of KOSΔPstLAT and rescue KOSΔPstR viruses in human skin *in vivo*.** Viral titres ($\log_{10}$ PFU/implant ± SEM; each data point represents an individual replicate) recovered from *in vivo* skin xenografts infected with 1,000 PFU KOSΔPstLAT and KOSΔPstR at 7 DPI (n = 5). In this assay, we could not capture the exact values for four datapoints (one of KOSΔPstLAT and three of KOSΔPstR) because their titres were higher than the set detection ability in the particular experiment. Therefore, the datapoints in the figure are set at the maximum value that could have been possible to detect (circles filled with grey colour). No statistical analysis was done without the exact values.
(TIF)

**S4 Fig. The glycoprotein gC expression by 17*syn*[+] and 17-37.** Western blot of gC in Vero cells infected with 17*syn*[+] and 17-37 (MOI 1) at 24 HPI. 17-37 expresses a truncated form of gC.
(TIF)

**S1 Data. The numerical data used in all figures are included in S1 Data except for the genome sequencing, which can be found in S1 Table.**
(XLSX)

## Acknowledgments

We thank David Bloom, University of Florida, for 17ΔPst and 17ΔPstR viruses and advice; David Bloom and Sanae Nakayama, University of Florida, for 17dmiR-H1/H6 and 17REd-miR-H1/H6 viruses; Don Coen, Harvard, for comments; Roger Everett, University of Glasgow, for 17*syn*[+] virus; Nigel Fraser, University of Pennsylvania, for 17ΔN/H virus; Paul Khavari, Stanford, for primary human keratinocytes; David Knipe, Harvard, for KOSΔPstLAT and KOSΔPstR viruses; David Leib, Dartmouth, for 17-37 BAC; Todd Margolis, Washington University, for pATD19 plasmid; Vida Shokoohi, Stanford Functional Genomics Facility, for

RNA-seq technical assistance; and Robert West, Stanford University, for LCM Smart-3SEQ advice.

## Author Contributions

**Conceptualization:** Emilia A. H. Vanni, Ann M. Arvin.

**Formal analysis:** Emilia A. H. Vanni, Joseph W. Foley, Andrew J. Davison.

**Funding acquisition:** Andrew J. Davison, Jennifer Moffat, Ann M. Arvin.

**Methodology:** Emilia A. H. Vanni, Joseph W. Foley, Andrew J. Davison, Marvin Sommer, Dongmei Liu, Phillip Sung, Jennifer Moffat, Leigh Zerboni, Ann M. Arvin.

**Writing – original draft:** Emilia A. H. Vanni, Ann M. Arvin.

**Writing – review & editing:** Emilia A. H. Vanni, Joseph W. Foley, Andrew J. Davison, Marvin Sommer, Dongmei Liu, Phillip Sung, Jennifer Moffat, Leigh Zerboni, Ann M. Arvin.

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
