## [Decision Letter · Decision Letter 0]

5 Oct 2020

Dear Dr Vanni,

Thank you very much for submitting your manuscript "The Latency-Associated Transcript Locus of Herpes Simplex Virus 1 is a Virulence Determinant in Human Skin" for consideration at PLOS Pathogens. As with all papers reviewed by the journal, your manuscript was reviewed by members of the editorial board and by several independent reviewers. The reviewers appreciated the attention to an important topic. Based on the reviews, we are likely to accept this manuscript for publication, providing that you modify the manuscript according to the review recommendations.

Specifically, please pay attention to issues brought up by all three reviewers in regards to inconsistencies in the body of the results for Figure 3 versus the Figure legend. If all of the items are addressed as detailed by the reviewers, I hope to make a final decision without sending the manuscript out for a final review.

Sincerely,

Paul D. Ling, Ph.D.

Associate Editor

PLOS Pathogens

Blossom Damania

Section Editor

PLOS Pathogens

Kasturi Haldar

Editor-in-Chief

PLOS Pathogens

orcid.org/0000-0001-5065-158X

Michael Malim

Editor-in-Chief

PLOS Pathogens

orcid.org/0000-0002-7699-2064

Reviewer Comments (if any, and for reference):

Reviewer's Responses to Questions

**Part I - Summary**

Reviewer #1: Figures 3D-G are numbered correctly in figures and legends but incorrectly in the body of the Results.

Please clarify that the significant results at 4dpi which become insignificant at 7 (?not 8) dpi are due to lack of residual keratinocytes destroyed by cytopathic spread of wt virus?

Figure 3E adds little as the results are not significant for the reasons explained. Either further replicates are needed for even nonparametric statistics or the results deleted or presented as supplementary figures. The two other LATS mutants are sufficiently convincing.

Reviewer #2: The revised manuscript, titled “The Latency-Associated Transcript Locus of Herpes Simplex Virus 1 is a Virulence Determinant in Human Skin”, reports that HSV-1 viruses mutated to interfere with viral transcription in the LAT locus can’t spread in human skin xenografts and cultured skin explants. The observation is interesting and potentially significant in the HSV-1 pathogenesis. The careful analysis and comprehensive comparison on wide-type and mutant viral genomes is to be praised in the detailed characterization of virus phenotypes.

Prior to this study, LAT has not been implied in the lytic infection of HSV life cycle. The authors showed that deletions of the upstream of LAT promoter and 5’ LAT region, as depicted in the strain 17ΔN/H, severely impaired HSV replication and lesion formation in skin. However, this defect in virus spread was caused neither by the defect in 2kb LAT production, which is the hall mark of HSV latency, nor by many nonsynonymous mutations found in the viral coding regions. Based on detailed mapping on the transcripts produced in the LAT locus, it seems like the 0.7 kb transcript upstream of LAT promoter and/or the 5’ exon sequences in the mLAT could be responsible for the observed phenomenon. The author added KOSΔPST mutant and confirmed the defective phenotype. But the mutation in ΔPST overlaps with 0.7kb transcript, so as 17ΔN/H. Since the 0.7kb transcript has been suggested in the virulence of HSV-1 17syn+, further evaluation using mutants separating in the two regions might be needed in providing clarity on the role of LAT in HSV-1 skin pathogenesis.

Reviewer #3: The authors have made significant improvements to this paper, which now provides compelling evidence that the HSV-1 latency associated transcripts act as a virulence determinant in human skin. The inclusion of additional LAT mutant viruses and, where available, revertant viruses strengthens their message. The inclusion of full sequence data for the mutant viruses will also act as a wake-up call to research groups working in this complex field. This is an important body of work that adds important new information concerning the role of LATs in pathogenesis outside of known functions in neuronal cells.

The authors should carefully review Figure 3, and the text describing this Figure in the results - it appears that some panels are miss-labelled/missing.

**Part II – Major Issues: Key Experiments Required for Acceptance**

Reviewer #1: (No Response)

Reviewer #2: (No Response)

Reviewer #3: (No Response)

**Part III – Minor Issues: Editorial and Data Presentation Modifications**

Reviewer #1: (No Response)

Reviewer #2: There is mixed use of LATs with LAT locus in the Author Summary, starting on line 44. Since the LAT splicing mutant have no effect in skin virulence, plus it is unknown whether other transcripts in the LAT promoter region express in the latency, it is more precise to use LAT locus than LATs, to avoid misleading.

Fig. 3C, Fig. 3D, 3E, 3F, 3G, seem shifted, the text and figure don’t match. There is one panel missing in Fig. 3.

Line 357-359 needs modification. Fig. 1 and 2C didn’t verify ICP0 retained its ubiquitin ligase function. Fig.2C only showed ICP0 expression.

Line 411-412 needs modification. Three references (47, 49, 50) only used 17 and KOS strain in cell culture. No mention on F strain.

Reviewer #3: The authors should carefully review Figure 3, and the text describing this Figure in the results - it appears that some panels are miss-labelled/missing.

PLOS authors have the option to publish the peer review history of their article (what does this mean?). If published, this will include your full peer review and any attached files.

Reviewer #1: No

Reviewer #2: No

Reviewer #3: No
---

## [Editor Report · Decision Letter 1]

12 Nov 2020

Dear Dr Vanni,

We are pleased to inform you that your manuscript 'The Latency-Associated Transcript Locus of Herpes Simplex Virus 1 is a Virulence Determinant in Human Skin' has been provisionally accepted for publication in PLOS Pathogens.

Best regards,

Paul D. Ling, Ph.D.

Associate Editor

PLOS Pathogens

Blossom Damania

Section Editor

PLOS Pathogens

Kasturi Haldar

Editor-in-Chief

PLOS Pathogens

orcid.org/0000-0001-5065-158X

Michael Malim

Editor-in-Chief

PLOS Pathogens

orcid.org/0000-0002-7699-2064
---

## [Editor Report · Acceptance letter]

17 Dec 2020

Dear Dr Vanni,

We are delighted to inform you that your manuscript, "The Latency-Associated Transcript Locus of Herpes Simplex Virus 1 is a Virulence Determinant in Human Skin," has been formally accepted for publication in PLOS Pathogens.

Best regards,

Kasturi Haldar

Editor-in-Chief

PLOS Pathogens

orcid.org/0000-0001-5065-158X

Michael Malim

Editor-in-Chief

PLOS Pathogens

orcid.org/0000-0002-7699-2064